# MODELING LABEL SPACE INTERACTIONS IN MULTI-LABEL CLASSIFICATION USING BOX EMBEDDINGS

**Dhruvesh Patel, Pavitra Dangati, Jay-Yoon Lee, Michael Boratko, Andrew McCallum**
Manning College of Information & Computer Sciences
University of Massachusetts Amherst
{dhruveshpate, sdangati, jaylee, mboratko, mccallum}@cs.umass.edu

## ABSTRACT

Multi-label classification is a challenging structured prediction task in which a set of output class labels are predicted for each input. Real-world datasets often have taxonomic relationships between labels which can be explicit, implicit, or partially observed. Most existing multi-label classification methods either ignore the label taxonomy or require the complete specification of the taxonomy at training and inference time to enforce coherence in their predictions. In this work we introduce the multi-label box model (MBM), a multi-label classification method that combines the encoding power of neural networks with the inductive bias of probabilistic box embeddings (Vilnis et al., 2018), which can be understood as trainable Venn-diagrams based on hyper-rectangles. By representing labels as boxes, MBM is able to capture taxonomic relations among labels without them being provided explicitly. Furthermore, since MBM learns the label-label relationships from data and represents them as calibrated conditional probabilities, it provides a high degree of interpretability. This interpretability also facilitates the injection of partial information about label-label relationships into model training, to further improve its consistency. We provide theoretical grounding for our method and show experimentally the model's ability to learn the true latent taxonomic structure from data. Through extensive empirical evaluations on twelve multi-label classification datasets, we show that MBM can significantly improve taxonomic consistency while maintaining the state-of-the-art predictive performance.[1]

## 1 INTRODUCTION

Multi-label classification is a machine learning task in which an input is associated with multiple categories. Many real-world multi-label classification datasets in modalities such as text categorization (Lewis et al., 2004), image classification (Lin et al., 2015; Krishna et al., 2016), entity typing (Murty et al., 2018; Onoe et al., 2021), functional genomics (Barutcuoglu et al., 2006; Clare, 2003), and so on, have a rich inter-dependent label structure that can be expressed using a taxonomy graph or a hierarchy. To be useful in practice, a model should produce predictions that are coherent with respect to the label taxonomy. For example, if a book is classified as *drama*, it should also be classified as *fiction* according to the label taxonomy of book genres in the left-hand side of Figure 1. More formally, we are given a label taxonomy in the form of a directed acyclic graph $G = (\mathcal{L}, \mathcal{T})$, where $(a, b) \in \mathcal{T}$ if and only if $a$ is a parent of $b$ in the taxonomy. A model will assign scores $s_\ell(x)$ for each label $\ell \in \mathcal{L}$. We say the scores are *coherent* with respect to the taxonomy if, for all edges $(a, b) \in \mathcal{T}$, $s_a(x) \geq s_b(x)$, and the *model* is consistent if this is the case for all inputs $x$. In the case of the book genre classification example, this implies that $s_{\text{fiction}}$ must be greater than or equal to $s_{\text{drama}}$, regardless of the input.

The problem of producing coherent predictions for multi-label classification has garnered a lot of attention in the machine learning literature (Wehrmann et al., 2018a; Giunchiglia & Lukasiewicz, 2020; Murty et al., 2018; Cerri et al., 2014). Most methods proposed to improve the coherence in predictions require the complete label taxonomy at inference time, and occasionally at training time

---

[1]The code and implementation details are available at `https://github.com/iesl/box-mlc-iclr-2022`

Figure 1: Left: A label taxonomy represented as a DAG. Center: A 2-dimensional box embedding of this taxonomy. Right: Scores assigned by the box embedding model to each label for two inputs.

as well, making these models hard to scale to large label spaces (Giunchiglia & Lukasiewicz, 2020; Wehrmann et al., 2018b). This brings forth a question: Can we utilize representation learning to model the label-label relationships implicitly in the embedding space?

Vilnis et al. (2018) introduced probabilistic box embeddings, which represent concepts as high dimensional hyper-rectangles, and demonstrated that they can embed DAGs efficiently using explicit information about the edges. Box embeddings represent edges by box-box containment as shown in Figure 1b. Representing the input and output labels in the same geometric space of boxes allows the multi-label taxonomy to be learned without an explicit taxonomic training signal. Moreover, there exists a large space of possible configurations that represent the same taxonomy, and if the label embeddings in the model gets close to any such configuration, then the model will always produce classifications that are coherent w.r.t the taxonomy, regardless of the input. We show through empirical evidence this is the case, and provide a formal proof for latter.

In this work, we propose the multi-label box model (MBM) that utilizes the geometry and probabilistic semantics of box embeddings to model label-label interactions in multi-label classification. MBM represents labels as boxes using free parameters and uses a deep neural network to embed the inputs in the same space. We perform coherence analysis of the model using two measures and show that MBM not only achieves state-of-the-art predictive performance but it also significantly improves the coherence of predicted scores w.r.t latent label taxonomy. Our analysis further shows that it is possible to retrieve the latent label-label relationships solely by analysing the learnt label representations inside the MBM, endowing the model with high degree of interpretability. Finally, we also present a way to utilize the interpretability of MBM to inject partial information about label-label relationships into the model thereby improving the coherence even further.

## 2 RELATED WORK

Multi-label classification tasks that exhibit strong label space structure in the form of explicit label taxonomy are termed hierarchical multi-label classification (HMLC) in machine learning literature. Most approaches for such tasks make use of the complete hierarchy at training time. These approaches can be categorized into two buckets (Silla & Freitas, 2010): (1) *Local approaches* that focus on local information for each label or clusters of labels in the hierarchy and classify them independently (Cerri et al., 2014; Huang et al., 2019), and (2) *Global approaches* that treat the problem as a structured classification task and take global interactions into account (Belanger & McCallum, 2016). In the most general setting, however, both local and global interactions between labels exist. The recent advances in deep learning (Wehrmann et al., 2018a) propose a specialized neural network architecture called Hierarchical Multi-Label Classification Network (HMCN-R and HMCN-F) that takes into account both local and global interactions by creating an ensemble of classifiers that can be trained using end-to-end gradient based training. However, HMCN does not try to enforce coherence strongly, focusing solely on predictive performance. In order to improve prediction coherence, recent works employ special loss functions on top of a neural network classifier to enforce coherence w.r.t the label taxonomy (Murty et al., 2018; Giunchiglia & Lukasiewicz, 2020). While effective, these approaches still use the label taxonomy explicitly, making them difficult to scale to very large label spaces.

Recent advances in representation learning provide various methods to embed large graphs and taxonomies parsimoniously in non-euclidean spaces. The most prominent of these embedding methods include hyperbolic embeddings (Nickel & Kiela, 2017; Ganea et al., 2018a;b) and box embedding (Vilnis et al., 2018; Dasgupta et al., 2021a). The use of representations other than Euclidean vectors for improving the coherence of multi-label classification has been limited to specific domains like text (Chatterjee et al., 2021) or specific tasks like entity typing (Onoe et al., 2021). Moreover, while both hyperbolic and box embedding can model hierarchical relationships, it has been shown that the box embedding can also model more general graphs like DAGs much more efficiently than hyperbolic embeddings (Patel et al., 2020; Boratko et al., 2021; Dasgupta et al., 2021b). Hence, we propose a model that uses box embedding to capture general label-label relationships without the explicit use of label taxonomy to improve the coherence of model predictions.

## 3 OVERVIEW OF BOX EMBEDDINGS

**Notations:** In the problem of multi-label classification, we are given a set of labels $\mathcal{L}$ where $L = |\mathcal{L}|$, and an instance can be labeled with an element $s \in \{0, 1\}^L$, where projection to the $i$th coordinate $\pi_i(s) = 1$ means that the $i$th label is true. We call the set of all such labelings $S$, and the associated probability space $(S, \mathcal{P}(S), P_S)$. We use $\mathbb{I}$ to denote the set of all finite closed intervals $[\mu^-, \mu^+]$ in $\Omega \subset \mathbb{R}$ plus the empty set, i.e. $\mathbb{I} := \{[\mu^-, \mu^+] \subset \Omega \mid \mu^+ \geq \mu^-\} \cup \emptyset$. We denote the smallest $\sigma$-algebra containing $\mathbb{I}$ as $\sigma(\mathbb{I})$ and, given a valid finite measure $\nu$, we consider the measure space $(\Omega, \sigma(\mathbb{I}), \nu)$. As a high dimensional generalization, $\mathbb{I}^d$ will denote a $d$-dimensional Cartesian product of $\mathbb{I}$.

**Definition 1** (Box Embedding (Vilnis et al., 2018)). Let $B : \mathbb{I}^d \to S$ be a measurable function such that $B^{-1} \circ \pi_i^{-1}(1) = \prod_i^d [\mu_i^-, \mu_i^+] \in \mathbb{I}^d$. A *box embedding* is defined as the function $\text{Box} : \mathcal{L} \to \mathbb{I}^d$ which maps a label $\ell \in \mathcal{L}$ to $B^{-1} \circ \pi_\ell^{-1}(\{1\}) \in \mathbb{I}^d$.

The definition of *box embeddings* induces a push-forward measure $Q$ on $S$ such that for any $R \subseteq S$, $Q(S) = \nu \circ B^{-1}(R)$. The complete joint probability distribution over the labels can be modeled using $Q$ as defined above; however, computing $B^{-1}(R)$ requires the use of inclusion-exclusion principle and hence is intractable for a general $R$.

In order to avoid local identifiability issues in training, Dasgupta et al. (2020) interpret $\mu_i^-$ (resp. $\mu_i^+$) as the location parameters of random variables $M_i^-$ (resp. $M_i^+$) that are distributed according to GumbelMax (resp. GumbelMin) distributions, leading to a meta-probabilistic generalization of box embedding which they call *Gumbel Box Process*. Since GumbelMax (resp. GumbelMin) is a max (resp. min) stable distribution, it enables the computation of the location parameters of the intersection box as given in the following definition.

**Definition 2** (Intersection Box (Dasgupta et al., 2020)). Let $A = \prod_{i=1}^d [a_i^-, a_i^+]$ and $B = \prod_{i=1}^d [b_i^-, b_i^+]$ be two gumbel boxes expressed using their location parameters, then the location parameters of the intersection of these two *gumbel boxes* are given as

$$A \tilde{\cap} B = \prod_{i=1}^d \left[ \beta \operatorname{lse}\left( \frac{a_i^-}{\beta}, \frac{b_i^-}{\beta} \right) , \ - \beta \operatorname{lse}\left( -\frac{a_i^+}{\beta}, -\frac{b_i^+}{\beta} \right) \right], \tag{1}$$

where $\operatorname{lse}(x, y) = \log(\exp(x) + \exp(y))$.

The expected volume of Gumbel boxes involves the Bessel Function of the Second Kind, however, as shown in Dasgupta et al. (2020), this integral can be reasonably approximated using softplus function leading to the following definition for *approximate bessel volume*.

**Definition 3** (Approximate Bessel Volume (Dasgupta et al., 2020)). For a gumbel box $B = \prod_{i=1}^d [b_i^-, b_i^+]$ we define the approximate Bessel volume $\lambda : \mathbb{I}^d \to \mathbb{R}_+$ as

$$\lambda(B) := \prod_{i=1}^d \beta \log\left( 1 + \exp\left( \frac{b_i^+ - b_i^-}{\beta} - 2\gamma \right) \right).$$

In the next section, we formally demonstrate the suitability of box embeddings for capturing taxonomic label relationships, and for that we first state a couple of useful facts regarding the Gumbel intersection and Bessel approximate volume.

**Proposition 1.** Approximate bessel volume *is monotonic with respect to set containment. That is for two Gumbel boxes $A, B$,*

$$a_i^- \geq b_i^- \quad and \quad a_i^+ \leq b_i^+, \quad \forall i \in \{1, \ldots, d\} \iff \lambda(A) \leq \lambda(B). \quad (2)$$

*Proof.* Follows from the monotonicity of $\log(1 + \exp(.))$. □

**Proposition 2.** *For any two Gumbel boxes $A, B$, $\lambda(A \tilde{\cap} B) \leq \lambda(B)$.*

*Proof.* The fact that $\max(x, y) \leq \text{lse}(x, y)$, and the statement of proposition 1 together imply the desired result. □

Since $\lambda$ is neither normalized nor additive, it cannot be used as a probability measure on $(\Omega, \sigma(\mathbb{I}^d))$. However, we can use proposition 1 and 2 to define a conditional probability model as follows.

**Corollary 1.** *For two gumbel boxes $A, B$, let $P_{\text{Box}}(A \mid B) = \frac{\lambda(A \tilde{\cap} B)}{\lambda(B)}$, then*

(i) *For any two gumbel boxes $A, B$, we have $0 \leq P_{\text{Box}}(A \mid B) \leq 1$.*

(ii) *$P_{\text{Box}}(A \mid C) \leq P_{\text{Box}}(B \mid C)$ for any three gumbel boxes $A, B, C$, with $a_i^- \geq b_i^-, a_i^+ \leq b_i^+$.*

## 4 MULTI-LABEL BOX MODEL

In order to perform the task of multi-label classification we need to model the conditional probabilities $P(Y|X)$ where $Y \in S$ and $X$ is the input. Using definition 1, we define label *box embeddings* $\text{Box}_\psi : \mathcal{L} \to \mathbb{I}^d$ as

$$\text{Box}_\psi(\ell_i) := \prod_{j=1}^d [\psi_{i,j}^-, \ \psi_{i,j}^- + \log(1 + \exp \psi_{i,j}^+)],$$

where $\psi^-, \psi^+ \in \mathbb{R}^{L \times d}$ are trainable parameters. The input instance $X$ is encoded as a fixed-width element of $\mathbb{I}^d$ using a parametric *instance box embedding* $\text{Box}_\theta = I^d \circ \mathcal{F}_\theta : \mathcal{X} \to \mathbb{I}^d$, where $\mathcal{F}_\theta : \mathcal{X} \to \mathbb{R}^d$ is a neural network with parameters $\theta$ and $I^d : \mathbb{R}^d \to \mathbb{I}^d$ defined as

$$I^d(x) := \prod_{i=1}^d [x_i - \delta, x_i + \delta],$$

where $\delta = 10^{-5}$. The conditional probability for $Y \in S$ given input $X$ is computed using conditional probability under the Gumbel box model as

$$P_{\text{MBM}}(Y|X; \psi, \theta) = \prod_{i=1}^L P_{\text{MBM}}(Y_i|X, \psi, \theta) := \prod_{i=1}^L P_{\text{Box}}(B^{-1} \circ \pi_i^{-1}(\{Y\}) \mid \text{Box}_\theta(X))$$

Using the definition of $P_{\text{Box}}$ as stated through corollary 1, we get the following expression for the conditional probability of $Y_i$ under the model, where the intersection $\tilde{\cap}$ is the *Gumbel Intersection* and measure $\lambda$ is *Approximate Bessel Volume*.

$$P_{\text{MBM}}(Y_i = 1|X; \psi, \theta) = \frac{\lambda\left(\text{Box}_\psi(\ell_i) \tilde{\cap} \text{Box}_\theta(X)\right)}{\lambda(\text{Box}_\theta(X))}$$

### 4.1 MODELING LABEL-SPACE INTERACTIONS

In Section 1, we alluded to the fact that the inductive bias of the MBM allows it to efficiently model partially specified first-order label interactions. Now we make this remark more concrete. If the partial specification of label interaction is defined using a taxonomy that can be represented as a directed acyclic graph (DAG), the following proposition shows that MBM has a strong inductive bias towards maintaining coherence in its scores.

**Proposition 3.** *Let $G = (\mathcal{L}, \mathcal{T})$ denote a DAG defined over the labels where $\mathcal{L}$ is the set of all labels and $\mathcal{T} = \{(\ell_i, \ell_j) \mid \ell_i, \ell_j \in \mathcal{L}, P_D(y_i = 1 \mid y_j = 1) = 1\}$ is the set of edges. Then there exists some $\psi$ such that $P_{(\psi, \theta)}(y_i = 1 \mid x) \geq P_{(\psi, \theta)}(y_j = 1 \mid x)$, for all $x, \theta$.*

*Proof.* For all $(\ell_i, \ell_j) \in \mathcal{T}$, let $\psi$ be such that $\text{Box}_\psi(\ell_j) \subseteq \text{Box}_\psi(\ell_i)$. Note that such $\psi$ exists since for each $i \in \{1, \ldots, L\}$, $\text{Box}_\psi(\ell_i)$ is defined using only $\psi_i$. It follows from corollary 1 that $P_{\text{Box}}(\text{Box}_\psi(\ell_i) \mid \text{Box}_\theta(X)) \geq P_{\text{Box}}(\text{Box}_\psi(\ell_j) \mid \text{Box}_\theta(X))$ for any $X, \theta$. □

## 4.2 LEARNING

The entire MBM is specified using parameters $(\psi, \theta)$ where $\psi \in \mathbb{R}^{2d \times L}$ are the label embedding parameters and $\theta$ are the parameters of the instance encoder neural network $\mathcal{F}_\theta$. Given data $D = \{(x^{(1)}, y^{(1)}), \ldots, (x^{(N)}, y^{(N)})\}$, the model parameters are learnt by minimizing negative log-likelihood loss

$$L_{\text{nll}}(\psi, \theta; D) = -\sum_{i=1}^{D} \sum_{j=1}^{L} \log P(y_j^{(i)} \mid x^{(i)}; \psi, \theta), \tag{3}$$

using the ADAM optimizer (Kingma & Ba, 2017). In order to empirically verify the intuition behind proposition 3, we also propose the use of label interaction loss

$$L_{\text{G}}(\psi) = -\sum_{(\ell_i, \ell_j) \in \mathcal{T}} s(\ell_i, \ell_j) + \sum_{(\ell_i, \ell_j) \notin \mathcal{T}} s(\ell_i, \ell_j) \tag{4}$$

that utilizes the geometry of box embeddings to inject partial information about label interactions specified using a label taxonomy $G = (\mathcal{L}, \mathcal{T})$. When label interaction loss is applied (MBM-T), the total loss is $L_T = L_{\text{nll}} + \nu L_G$, where $\nu$ is a hyperparameter. For the Box model, label interaction score for a pair of labels is defined as

$$s_{\text{MBM}}(\ell_i, \ell_j) := \log P_{\text{Box}}(\text{Box}_\psi(\ell_i) \mid \text{Box}_\psi(\ell_j)). \tag{5}$$

## 5 BASELINES

Our choice of baselines reflects the focus of this work, i.e., introducing coherence in prediction using suitable representation spaces. To this end, our baselines consist of a high-performing neural network that only uses Euclidean vector representations (MVM), and another that uses hyperbolic representations (MHM). In order to test the importance of the probabilistic semantics used to formulate MBM, we also include as baseline, a non-probabilistic box model as defined in Abboud et al. (2020). The base input encoder architecture $\mathcal{F}_\theta$ in all these baselines is same as the one used in MBM.

**Multi-label Vector Model (MVM)** An input encoder neural network $\mathcal{F}_\theta : \mathcal{X} \to \mathbb{R}^d$ is used to encode the inputs and a label embedding matrix $\psi$ is used to represent the labels. The conditional probability of labels given the input and label interaction scores are given as

$$P_{\text{MVM}}(y_l = 1 \mid x; \psi, \theta) := \sigma(\mathcal{F}_\theta(X)^T M_\theta \, \psi_l), \quad \text{and} \quad s_{\text{MVM}}(l_i, l_j) := \sigma(\psi_{l_i}^T M_\theta \, \psi_{l_j}),$$

respectively, where $\sigma$ is the logistic sigmoid function, and $G_\theta \in \mathbb{R}^{d \times d}$ is a matrix of trainable parameters. The parameters $(\theta, \psi)$ are learnt through $L_T$ (Eq. 3,4). Note that, when label interaction is not being explicitly modeled i.e., $\nu = 0$, $M$ is fixed as the identity matrix and MVM reduces to a special case of multi-layer perceptron.

**Multi-label Hyperbolic Model (MHM)** As discussed in Chatterjee et al. (2021), the isometry between the Lorentz model and Poincare disk model for hyperbolic geometry can be used produce the retraction formula used to project $d$-dimensional euclidean vectors to $d$-dimensional Poincaré ball $\mathcal{B}^d = \{x \in \mathbb{R}^d \mid \|x\| < 1\}$. Here the projection operator $\Pi : \mathbb{R}^d \to \mathcal{B}^d$ and the distance $d : \mathcal{B}^d \times \mathcal{B}^d \to \mathbb{R}_+$ in the hyperbolic space are given as:

$$\Pi(x) := \frac{x}{1 + \sqrt{1 + \|x\|_2^2}} \quad , \qquad d(u, v) := \text{arcosh}\left(1 + 2\frac{\|u - v\|}{(1 - \|u\|_2^2)(1 - \|v\|_2^2)}\right).$$

Specifically, the input is first encoded using $\mathcal{F}_\theta$ and then projected into $\mathcal{B}^d$. The unnormalized score for each label is computed as the negative of the distance between the hyperbolic projections of encoded input and label representation. Since the hyperbolic distance function consists of arcosh, the negative distance is interpreted as log-probability score. Hence, the conditional log-probability of the labels and the label interaction score is given, respectively, by the following expressions, which are then used to learn the parameters $(\psi, \theta)$ in the same way as MVM and MBM.

$$\log P_{\text{MHM}}(y_l = 1 \mid x) = -d(\Pi(\mathcal{F}_\theta(x)), \Pi(\psi_l)) \quad \text{and} \quad s_{\text{MHM}}(l_i, l_j) := -d(\Pi(\psi_{l_i}), \Pi(\psi_{l_j})),$$

**Non-probabilistic box model (BoxE)**: In order to test the importance of the probabilistic semantics used to formulate the MBM, we also include as baseline, a box model that does not use probabilistic semantics. For this, we replace the $P_{\text{MBM}}$ with the non-probabilistic score defined in Abboud et al. (2020) and replace the log-likelihood loss ($L_{\text{nll}}$) with a margin based loss $L_{\text{margin}}$. [1]

## 6 EVALUATION AND RESULTS

We evaluate the performance of MBM using 12 real-world multi-label classification datasets (Dimitrovski et al., 2011; Clare, 2003) spanning across three domains: text (Enron), images (Imclef07a, Imclef07d, Diatoms), and functional genomics (Expr, Cellcycle, Derisi, Spo) . These datasets are the ideal test bed as they provide explicit label taxonomies with different connectivity: trees, forests, and more general DAGs. Moreover, all the labels of all training and test instances respect the label taxonomy. The datasets include both categorical and continuous input features. We convert the categorical features into one-hot feature vectors and standardize all continuous features. The input encoder $\mathcal{F}_\theta$ uses a common architecture for all models consisting of an MLP with a maximum of 3 layers. We perform a grid search over number of MLP layers, activation function, hidden dimensions, dropout, learning rate and use the best parameters for each model.[2] The Mean Average Precision (MAP), that is the mean over the average precision values across instances in the test set, is used to evaluate the predictive performance of the models. Table 1 reports average metric values across 10 runs with different random seeds (the standard error intervals are small and are omitted for the sake of readability). As seen in Table 1, the predictive performance (MAP) of MBM is better than all other embedding based methods on at least 11 out of 12 datasets. To check the statistical significance of our results, following Demšar (2006), we first perform the Friedman test and obtain a p-value of $2 \times 10^{-9}$. The critical diagram of the post-hoc Nemenyi test (Figure 2a) performed after the Friedman test allows us to conclude that difference in predictive performance between MBM and all other embedding based models is statistically significant. Furthermore, from these results, one can conclude with confidence that MBM with its probabilistic formulation preforms significantly better than BoxE, which is the non-probabilistic variant of box embeddings. In the following section, we discuss another important aspect of performance, that is, the coherence of predictions.

### 6.1 COHERENCE ANALYSIS

MAP is a good metric for predictive performance, however it does not take into account the inconsistencies in the predicted scores w.r.t. the label taxonomy. For instance, recalling the earlier example in Figure 1, a consistent model would always assign higher score to *fiction* when compared to *drama*, since a book classified as *drama* should also be classified as *fiction*. Since MAP is incapable of capturing such coherence conditions, we perform further analysis to check the coherence of the predicted scores. This is done by considering two opposite perspectives towards coherence evaluation. The first one is a punitive approach, where we count the number of instances that are given inconsistent scores w.r.t the taxonomy. In the second one, a more constructive approach, we try to check the amenability of the model, by applying post-hoc corrections to the predicted scores to enforce coherence. The results of these two approaches are presented by using Constraint Violation (CV) and Mean Average Precision post Coherence correction (CMAP), which are described below. Note that we do not recommend the use of either CV or CMAP as the sole metrics to evaluate the model, we present these as measures to analyse the coherence of the model predictions.

**Constraint violation** is a punitive measure that quantifies the extent to which the label scores generated by the model violate the partial ordering of the latent label taxonomy regardless of true labels for the instances. Hence, lower value of CV implies higher taxonomic coherence in the predictions.

$$\text{CV}(s) = \frac{1}{|D||\mathcal{T}|} \sum_{k=1}^{|D|} \sum_{(l_i, l_j) \in \mathcal{T}} \mathbb{1}\left(s_i^{(k)} - s_j^{(k)} < 0\right). \tag{6}$$

**Mean Average Precision post Coherence correction (CMAP)**: Given a complete or partial taxonomy $G = (\mathcal{L}, \mathcal{T})$ for the labels, coherence can be imposed post-hoc by applying a modification function $\delta : \mathbb{R}^L \to \mathbb{R}^L$ to the label-scores produced by the model such that $\delta(s_i) - \delta(s_j) < 0$ for

---

[2]Please see the *Reproducibility Statement* after Section 8, for further details on the implementation of the baselines and the experiments.

Table 1: Performance comparison of MBM models with the baselines for the 12 multi-label classification datasets. The left section compares the models that do not require explicit taxonomy, i.e., BoxE, MVM, MHM and MBM. In the left section, the models with the best performance w.r.t. MAP are highlighted. The right section shows the performance when we include taxonomy information in training through $L_G$ (MVM-T, MHM-T and MBM-T), where the highlighted cells indicate an improvement in performance (MAP and CV) w.r.t. the respective non-T model. All metrics reported are averaged across five runs with different seeds.

| Dataset | Metric | BoxE | MVM | MHM | MBM | MVM-T | MHM-T | MBM-T | C-HMCNN |
|---|---|---|---|---|---|---|---|---|---|
| ExprFUN | MAP↑ | 37.30 | 38.37 | 31.91 | 38.45 | 37.94 | 31.90 | 38.42 | 38.41 |
| | CMAP↑ | 37.92 | 37.66 | 32.05 | 38.72 | 37.41 | 32.02 | 38.67 | 38.41 |
| | CV↓ | 4.79 | 1.99 | 1.94 | 2.55 | 1.97 | 1.92 | 1.87 | 0 |
| CellcycleFUN | MAP↑ | 31.96 | 31.68 | 28.76 | 34.20 | 31.61 | 28.74 | 34.61 | 34.35 |
| | CMAP↑ | 32.70 | 31.34 | 28.92 | 34.39 | 31.33 | 28.89 | 34.78 | 34.35 |
| | CV↓ | 4.02 | 3.42 | 1.78 | 1.77 | 3.45 | 1.78 | 1.35 | 0 |
| DerisiFUN | MAP↑ | 26.66 | 23.70 | 26.87 | 28.69 | 24.16 | 24.40 | 28.71 | 28.19 |
| | CMAP↑ | 26.96 | 24.13 | 26.98 | 28.86 | 24.35 | 24.52 | 28.88 | 28.19 |
| | CV↓ | 2.27 | 4.79 | 0.95 | 1.67 | 4.01 | 0.85 | 1.43 | 0 |
| SpoFUN | MAP↑ | 27.97 | 25.18 | 26.58 | 30.10 | 24.21 | 26.57 | 29.62 | 29.18 |
| | CMAP↑ | 28.38 | 25.38 | 26.79 | 30.27 | 24.55 | 26.79 | 29.78 | 29.18 |
| | CV↓ | 2.75 | 4.23 | 1.68 | 1.75 | 4.73 | 1.69 | 1.53 | 0 |
| ExprGO | MAP↑ | 46.75 | 44.92 | 40.53 | 48.45 | 44.97 | 40.52 | 48.45 | 48.61 |
| | CMAP↑ | 47.28 | 41.78 | 40.71 | 48.56 | 41.84 | 40.70 | 48.56 | 48.61 |
| | CV↓ | 5.74 | 7.05 | 5.12 | 2.46 | 7.05 | 5.19 | 1.91 | 0 |
| CellcycleGO | MAP↑ | 43.08 | 44.25 | 39.77 | 44.28 | 44.19 | 39.74 | 44.93 | 45.61 |
| | CMAP↑ | 43.79 | 41.09 | 39.90 | 44.23 | 41.02 | 39.76 | 45.01 | 45.61 |
| | CV↓ | 5.06 | 3.07 | 2.35 | 2.84 | 3.03 | 2.49 | 2.16 | 0 |
| DerisiGO | MAP↑ | 40.44 | 41.22 | 40.16 | 42.03 | 41.13 | 40.10 | 42.02 | 42.24 |
| | CMAP↑ | 40.73 | 38.21 | 40.28 | 42.14 | 38.21 | 40.20 | 42.12 | 42.24 |
| | CV↓ | 3.16 | 3.43 | 1.98 | 2.37 | 3.46 | 2.02 | 1.13 | 0 |
| SpoGO | MAP↑ | 40.88 | 42.19 | 39.81 | 42.22 | 42.20 | 39.70 | 41.74 | 42.77 |
| | CMAP↑ | 41.27 | 38.96 | 39.89 | 42.31 | 39.04 | 39.77 | 41.54 | 42.77 |
| | CV↓ | 3.89 | 2.81 | 1.93 | 2.68 | 2.77 | 1.90 | 1.80 | 0 |
| Enron | MAP↑ | 80.44 | 73.68 | 75.95 | 79.95 | 73.68 | 75.62 | 80.06 | 80.04 |
| | CMAP↑ | 80.46 | 66.87 | 76 | 79.94 | 66.87 | 75.68 | 80.05 | 80.04 |
| | CV↓ | 0.20 | 2.53 | 0.29 | 0.04 | 2.53 | 0.36 | 0.03 | 0 |
| Diatoms | MAP↑ | 43.71 | 73.06 | 56.97 | 79.14 | 72.65 | 56.86 | 79.14 | 76.23 |
| | CMAP↑ | 45.16 | 72.44 | 56.14 | 79.30 | 72.18 | 56.07 | 79.23 | 76.23 |
| | CV↓ | 6.39 | 18.97 | 5.59 | 3.46 | 19.20 | 5.55 | 0.34 | 0 |
| Imclef07a | MAP↑ | 83.71 | 77.14 | 65.29 | 91.45 | 78.22 | 65.30 | 69.26 | 90.26 |
| | CMAP↑ | 84.73 | 76.56 | 66.01 | 91.73 | 77.46 | 66.01 | 69.48 | 90.26 |
| | CV↓ | 12.73 | 23.02 | 4.75 | 5.65 | 22.86 | 4.75 | 2.40 | 0 |
| Imclef07d | MAP↑ | 87.95 | 88.49 | 75.72 | 89.49 | 88.59 | 75.69 | 89.56 | 89.22 |
| | CMAP↑ | 88.93 | 86.89 | 76.98 | 89.99 | 86.87 | 76.95 | 90.07 | 89.22 |
| | CV↓ | 11.93 | 10.72 | 7.52 | 7.16 | 11.02 | 7.56 | 5.66 | 0 |
| **Avg. Rank ↓** | MAP | 4.92 | 5.29 | 6.42 | **2.08** | 5.38 | 7.33 | 2.42 | 2.17 |
| | CMAP | 4.08 | 6.08 | 6.17 | **1.96** | 6.25 | 6.92 | 2.29 | 2.25 |
| | CV | 6.67 | 7 | 3.83 | 4.33 | 7 | 4 | 2.17 | **1** |

all $(l_i, l_j) \in \mathcal{T}$. This can be achieved by modifying the score $s_i$ for each label $l_i$ to be either the maximum ($\delta_G^M$) of the scores of any of its descendants or the minimum of the scores of its ancestors ($\delta_G^m$) in the taxonomy $G$. Concretely, given the scores $s \in \mathbb{R}^L$ produced by a model, for an input, the two modification functions are given as

$$\delta_G^m(s)_i = \min_{l_j \in \mathrm{Anc}_G(l_i) \cup \{l_i\}} s_j, \qquad \delta_G^M(s)_i = \max_{l_j \in \mathrm{Des}_G(l_i) \cup \{l_i\}} s_j,$$

where $\mathrm{Anc}_G(l)$ and $\mathrm{Des}_G(l)$ is the set of ancestors and descendants, respectively, of $l$ in the graph $G$. In practice, if one is given a partial label taxonomy, one would select whichever post-hoc modifi-

cation performs better, and we refer to the maximum MAP obtained after applying one of the modification functions $\delta_G^m, \delta_G^M$ to the scores as CMAP.[3] Intuitively, a CMAP value close to MAP value implies that the model is more perceptive of the latent hierarchy in the label space. As seen from the critical diagram of the Nemenyi test (Figure 2b), when coherence is considered along with the predictive performance (CMAP), MBM outperforms MVM and MHM (with statistical significance) indicating that MBM is much more perceptive of the latent taxonomy compared to other embedding based models (MVM and MHM). In conclusion, while on one extreme there is MVM, which exhibits reasonable good predictive performance but fails to maintain coherence w.r.t the taxonomy, on the other extreme we have MHM that exhibits lowest constraint violations but gives inadequate predictive performance. MBM, however, demonstrates good characteristics on both fronts–predictive performance as well as coherence.

**Comparison with the state-of-the-art:** C-HMCNN (Giunchiglia & Lukasiewicz, 2020), is the current state-of-the-art method for hierarchical MLC, and includes the modification function $\delta_G^M$ (called MCM in their work) in their model directly, applying it to scores produced by a multi-layer perceptron at training as well as inference time.[4] At training time, this modification is accompanied by a clever modification of BCE loss to form a novel MCLoss which works well with the modification function during training. There are two key differences between C-HMCNN and MBM that one needs to keep in mind when comparing their performance. First, C-HMCNN does not attempt to use the label representations themselves to increase coherence—an idea central to MBM and this work. Second, unlike MBM, C-HMCNN requires the complete label taxonomy at training and inference time to enforce the coherence. Owing to the use of complete label taxonomy at inference time, C-HMCNN always has 0 constraint violations. However, as seen from the Table 1 and the Figure 2, MBM, which does not require the label taxonomy at all, performs comparably to C-HMCNN in terms of MAP (p-value of 0.677 in two sided Wilcoxon test between MBM and C-HMCNN shows that there is no statistical difference between their performances).

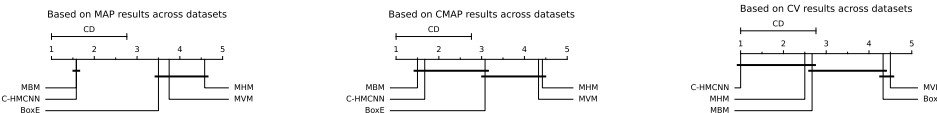

Figure 2: Critical diagrams of the post-hoc Nemenyi test across all 12 datasets.

# 7 ANALYSIS OF LEARNED LABEL EMBEDDINGS

In this section, we analyze the geometry of the learned label embeddings, finding that the simple geometry of box embeddings endows the MBM model with high degree of interpretability. In order to verify that label box embeddings are producing consistent scores by using inclusion in the box space, we inject into the model the taxonomy information through the additional loss term (Eq. 4). As seen from the right section in Table 1, injecting explicit taxonomic information into the label embeddings further reduces the constrain violation for MBM on all twelve datasets. However, the same does not aid the MVM model and even drops its predictive performance significantly. This underscores the importance of embedding geometry for inducing taxonomic coherence, and validates our intuition about the arrangement of label embeddings in MBM.

To determine the extent to which the label embeddings capture the latent label taxonomy without it being explicitly provided, we perform ancestor-descendant classification solely using the learned label embeddings. Each pair of labels $(l_i, l_j)$ get a score $\beta(i, j)$ that is determined using their corresponding label embeddings $\psi_i, \psi_j$. Since MVM and MBM have different geometrical interpretations, we use different scoring for each. Specifically, $\beta_{\mathrm{MVM}}(i, j) = \psi_i \cdot \psi_j / \|\psi_j\|$, $\beta_{\mathrm{MBM}}(i, j) = s_{\mathrm{MBM}}(l_i, l_j)$. These scores are then compared to true ancestor-descendant relations in the taxonomy to obtain respective ROC curves as shown in Figure 3. As seen in Table 3, MBM captures the true label taxonomy the best ($AUC \geq 0.87$) for all datasets.

---

[3]We find that $\delta_G^m$ produces lesser change in the scores predicted by all the models (MBM, MVM and MHM), and hence is used as the modification function for this analysis (see Appendix E for further discussion on this)

[4]Comparision with another competitive baseline HMCN is provided in Appendix D.

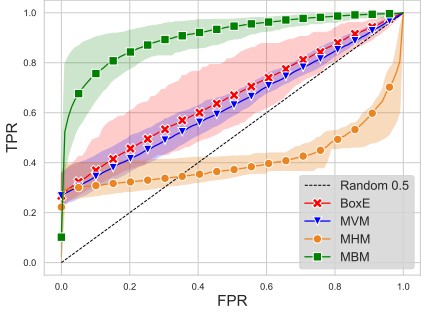

| Dataset | BoxE | MVM | MHM | MBM |
|---|---|---|---|---|
| ExprFUN | 0.59 | 0.61 | 0.40 | **0.89** |
| CellcycleFUN | 0.71 | 0.59 | 0.40 | **0.93** |
| DerisiFUN | 0.64 | 0.61 | 0.41 | **0.87** |
| SpoFUN | 0.65 | 0.61 | 0.39 | **0.87** |
| Enron | 0.77 | 0.66 | 0.49 | **0.92** |
| Diatoms | 0.67 | 0.68 | 0.44 | **0.96** |
| Imclef07a | 0.58 | 0.63 | 0.45 | **0.87** |

Figure 3: The figure on the left shows the envelope of the ROC curves for the ancestor-descendant relationship classification in the label space for different embedding based models across the datasets and the table on the right is the area under the ROC curves. As seen, MBM implicitly captures the label taxonomy well compared to all other models.

Furthermore, for hyperberbolic space, it is suggested that the magnitude of embeddings relate to the level of generality in taxonomy (Nickel & Kiela, 2017). We show that the same observation holds for box embeddings, with the vector embedding magnitude replaced by box embedding volume. To see this, we compute the Spearman rank correlation between the number of

Table 2: Spearman rank correlation between the number of descendants in the label taxonomy with each of the following: embedding magnitude for MVM, negative embedding magnitude for MHM and box embedding volume for MBM.

| Model | Expr FUN | Cellcycle FUN | Derisi FUN | Spo FUN | Enron | Diatoms | Imclef07a |
|---|---|---|---|---|---|---|---|
| MHM | -0.37 | 0.38 | 0.40 | 0.38 | 0.19 | 0.31 | 0.32 |
| MVM | -0.06 | -0.11 | -0.01 | 0.06 | -0.11 | 0.04 | -0.02 |
| MBM | 0.47 | 0.49 | 0.50 | 0.48 | 0.47 | 0.23 | 0.43 |

descendants of a node in the true taxonomy and the embedding magnitude, negative embedding magnitude, and embedding volume for MVM, MHM, and MBM, respectively. The correlation values reported in Table 2 confirm our intuition regarding box embeddings stated above.[5] It is interesting to note that as shown by the correlation between the vector magnitude and level of the label node as well as low constraint violations, the MHM model does capture the depth of the label nodes in the taxonomy. However, as shown by the ROC curves in Figure 3, unlike MBM, it fails to capture the exact connectivity of the nodes at different depths in the taxonomy.

## 8 CONCLUSION

In this work, we demonstrate that box embeddings with its probabilistic formulation can effectively capture taxonomic relations present between labels in multi-label classification without requiring explicit access to the taxonomy. The proposed model achieves a fine balance between predictive performance and coherence. Furthermore, we find that the taxonomic relationships between labels can be easily injected via extra supervision during training, increasing the coherence of the predictions further. Since the model has the same computational complexity as a simple neural network model, and unlike C-HMCNN it does not require the label taxonomy, the proposed model can be scaled to work with extremely large label spaces. Moreover, due to the flexibility of using any neural network encoder this model can be easily extended to different input modalities, such as raw text, images, etc. We wish to pursue these two directions in our future work.

## REPRODUCIBILITY STATEMENT

In this section, we provide pointers to information necessary to reproduce the results mentioned in this paper. The description of the datasets with various statistics, links to download them, and instructions to pre-process them are provided in Appendix B. The BoxE baseline is further described

---

[5]Refer to Appendix H for scatter plots of box embedding volume vs. the number of children in the label taxonomy.

with expressions for distance and the margin based loss function in Appendix A.1. The hyper-parameter settings for each model-dataset combination (all 96 settings) used to produce the reported results are provided in 3. We also include a discussion on computational complexity of the models in Appendix A.2. Finally, the code for all the models (including the baselines) used in this paper is available at `https://github.com/iesl/box-mlc-iclr-2022`.

## ACKNOWLEDGEMENTS

The authors would like to thank the members of the Information and Extraction Synthesis Laboratory (IESL) at UMass Amherst for helpful discussions. This work was partially supported by IBM Research AI through the AI Horizons Network and the Chan Zuckerberg Initiative under the project Scientific Knowledge Base Construction. Additional support was provided by the National Science Foundation (NSF) under Grant Numbers IIS-1763618, IIS-1922090, and IIS-1955567, the Defense Advanced Research Projects Agency (DARPA) via Contract No. FA8750-17-C-0106 under Sub-award No. 89341790 from the University of Southern California, and the Office of Naval Research (ONR) via Contract No. N660011924032 under Subaward No. 123875727 from the University of Southern California. The U.S. Government is authorized to reproduce and distribute reprints for Governmental purposes notwithstanding any copyright notation thereon. The views and conclusions contained herein are those of the authors and should not be interpreted as necessarily representing the official policies or endorsements, either expressed or implied, of IBM, CZI, NSF, DARPA, ONR, or the U.S. Government.

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

## A  IMPLEMENTATION DETAILS

In this section, we describe the implementation details: training procedure and hyper-parameter search.

**Frameworks used:** We implement all the models described in this work using PyTorch (Paszke et al., 2019). We also make use of NLP specific abstractions over PyTorch provided by AllenNLP (Gardner et al., 2017). We also make use of the abstractions provided in the Box Embedding library Chheda et al. (2021) in our implementation of MBM. For the extensive hyper-parameter search, we use the academic version of Weights & Biases library (Biewald, 2020).

**Data pre-processing:** The datasets were pre-processed in the exact same manner as in Giunchiglia & Lukasiewicz (2020), i.e., the categorical features were converted to one-hot feature vectors and the continuous input features were standardized. Also, following the Giunchiglia & Lukasiewicz (2020); Wehrmann et al. (2018a), the labels corresponding to the root nodes were removed from training and evaluation. Further details of the original datasets is provided in B.

**Training:** We used ADAM (Kingma & Ba, 2017) with a batch size of 4 to learn the model parameters for all the models. Since the naive implementation of the label interaction loss described in Eq. 4 is too expensive to compute at each mini-batch step, we approximate it by randomly sampling (without replacement), at each mini-batch step, a subset of edges $\tilde{\mathcal{T}} \sim \mathcal{T}$. The size of the sampled set is a hyper-parameter (last column in Table 3).

**Hyper-parameter search:** The following hyper-parameters were searched using the performance on the validation set: optimizer learning rate, hidden dimensions of the MLP, number of hidden layers (maximum of 3), activation functions, dropout probabilities, weight decay and sampling percentage for the labels for computing the label interaction loss($L_G$) for the MHM-T, MBM-T and MVM-T models. After an initial investigation, we identified that the softplus activation performed the best for all models and hence, the activation was fixed to softplus for the final grid search. Also note that in order to be comparable to the previous work that uses these datasets, we had to divide our experiments into two kinds of runs—the search runs and the final runs. First, in the *search runs*, we use a validation set to obtain the best hyper-parameters for each model-dataset combination. After identifying the best set of hyper-parameters the training and validation sets were combined. We call the models trained on this combined dataset as the *final runs*. Since one cannot use early stopping without a validation set, the number of epochs to train the models in the final runs was also obtained as a hyper-parameter and was set to the best epoch for the corresponding search run. All the results are reported using the metrics obtained in the final runs (10 runs with different random seeds for each dataset-model pair with best hyper-parameters obtained from the corresponding search runs). Table 3 summarizes the search ranges used and Table 4 presents the final hyper-parameters obtained.

Table 3: Summary of the hyper-parameter search ranges for each dataset and model. The best hyper-parameters for each model and dataset combination were picked using grid search using MAP on the validation set. Note that $\nu$ and *label sample percent* are only applicable to the MBM-T, MVM-T and MHM-T models.

| Datasets | batch size | lr | hidden dim | layers | activation | dropout | $L_G$ weight ($\nu$) | label sample percent |
|---|---|---|---|---|---|---|---|---|
| all | 4 | 1e-4, 5e-4, 1e-3, 1e-2 | 250, 500, 1000, 1750 | 1,2,3 | sigmoid, relu, softplus | 0.0, 0.1, 0.3, 0.5 | 1e-3, 1e-6, 1e-7, 1e-9 | 10,20,30 |

Table 4: Best hyper-parameters obtained through grid search for each dataset-model combination. It should be noted that the activation function chosen for all model configurations was softplus.

| Dataset | Model | learning rate | hidden dim | layers | dropout | % label sample | $\nu$ |
|---|---|---|---|---|---|---|---|
| CellcycleFUN | BoxE | 0.001 | 500 | 3 | 0.3 | - | - |
| CellcycleFUN | C-HMCNN | 0.001 | 500 | 3 | 0.3 | 100 | - |
| CellcycleFUN | MBM | 0.0001 | 1750 | 3 | 0 | - | - |
| CellcycleFUN | MBM-T | 0.0001 | 500 | 3 | 0 | 30 | 1e-07 |
| CellcycleFUN | MHM | 0.0001 | 1000 | 3 | 0 | - | - |
| CellcycleFUN | MHM-T | 0.0001 | 1000 | 3 | 0 | 30 | 1e-07 |
| CellcycleFUN | MVM | 0.001 | 1750 | 2 | 0 | - | - |
| CellcycleFUN | MVM-T | 0.001 | 1750 | 2 | 0 | 30 | 1e-07 |

**Table 4 continued from previous page**

| Dataset | Model | learning rate | hidden dim | layers | dropout | % label sample | $\nu$ |
|---------|-------|--------------|-----------|--------|---------|---------------|---|
| CellcycleGO | BoxE | 0.001 | 500 | 3 | 0.3 | - | - |
| CellcycleGO | C-HMCNN | 0.001 | 500 | 2 | 0.5 | 100 | - |
| CellcycleGO | MBM | 0.0001 | 1750 | 3 | 0 | - | - |
| CellcycleGO | MBM-T | 0.0001 | 1750 | 3 | 0 | 10 | 1e-07 |
| CellcycleGO | MHM | 0.001 | 1750 | 3 | 0.3 | - | - |
| CellcycleGO | MHM-T | 0.001 | 1750 | 3 | 0.3 | 30 | 1e-07 |
| CellcycleGO | MVM | 0.001 | 500 | 2 | 0.5 | - | - |
| CellcycleGO | MVM-T | 0.001 | 500 | 2 | 0.5 | 30 | 1e-07 |
| DerisiFUN | BoxE | 0.001 | 500 | 3 | 0.3 | - | - |
| DerisiFUN | C-HMCNN | 0.0001 | 1000 | 3 | 0.3 | 100 | - |
| DerisiFUN | MBM | 0.0001 | 500 | 3 | 0 | - | - |
| DerisiFUN | MBM-T | 0.0001 | 500 | 3 | 0 | 30 | 1e-07 |
| DerisiFUN | MHM | 0.001 | 500 | 3 | 0.1 | - | - |
| DerisiFUN | MHM-T | 0.001 | 500 | 3 | 0.1 | 30 | 1e-07 |
| DerisiFUN | MVM | 0.001 | 1750 | 3 | 0 | - | - |
| DerisiFUN | MVM-T | 0.001 | 1750 | 3 | 0 | 30 | 1e-07 |
| DerisiGO | BoxE | 0.001 | 500 | 3 | 0.3 | - | - |
| DerisiGO | C-HMCNN | 0.001 | 1000 | 3 | 0.3 | 100 | - |
| DerisiGO | MBM | 0.0001 | 500 | 3 | 0 | - | - |
| DerisiGO | MBM-T | 0.0001 | 500 | 3 | 0 | 20 | 1e-07 |
| DerisiGO | MHM | 0.001 | 500 | 3 | 0 | - | - |
| DerisiGO | MHM-T | 0.001 | 500 | 3 | 0 | 30 | 1e-07 |
| DerisiGO | MVM | 0.001 | 500 | 2 | 0.3 | - | - |
| DerisiGO | MVM-T | 0.001 | 500 | 2 | 0.3 | 30 | 1e-07 |
| Diatoms | BoxE | 0.001 | 1750 | 3 | 0.3 | - | - |
| Diatoms | C-HMCNN | 0.0001 | 500 | 3 | 0.1 | 100 | - |
| Diatoms | MBM | 0.001 | 1750 | 3 | 0 | - | - |
| Diatoms | MBM-T | 0.0001 | 1750 | 3 | 0 | 30 | 1e-07 |
| Diatoms | MHM | 0.0001 | 1750 | 3 | 0 | - | - |
| Diatoms | MHM-T | 0.0001 | 1750 | 3 | 0 | 30 | 1e-07 |
| Diatoms | MVM | 0.0001 | 1000 | 3 | 0 | - | - |
| Diatoms | MVM-T | 0.0001 | 1000 | 3 | 0 | 30 | 1e-07 |
| Enron | BoxE | 0.001 | 500 | 2 | 0.3 | - | - |
| Enron | C-HMCNN | 0.0001 | 500 | 2 | 0.3 | 100 | - |
| Enron | MBM | 0.0001 | 1750 | 2 | 0 | - | - |
| Enron | MBM-T | 0.0001 | 1750 | 2 | 0 | 30 | 1e-07 |
| Enron | MHM | 0.0001 | 1750 | 2 | 0.1 | - | - |
| Enron | MHM-T | 0.0001 | 1750 | 2 | 0.1 | 30 | 1e-07 |
| Enron | MVM | 0.0005 | 500 | 3 | 0 | - | - |
| Enron | MVM-T | 0.0005 | 500 | 3 | 0 | 30 | 1e-07 |
| ExprFUN | BoxE | 0.001 | 500 | 3 | 0.3 | - | - |
| ExprFUN | C-HMCNN | 0.0001 | 1000 | 3 | 0.3 | 100 | - |
| ExprFUN | MBM | 0.0005 | 1000 | 3 | 0 | - | - |
| ExprFUN | MBM-T | 0.0005 | 1000 | 3 | 0 | 30 | 1e-07 |
| ExprFUN | MHM | 0.0001 | 1750 | 3 | 0 | - | - |
| ExprFUN | MHM-T | 0.0001 | 1750 | 3 | 0 | 30 | 1e-07 |
| ExprFUN | MVM | 0.0005 | 500 | 3 | 0.5 | - | - |
| ExprFUN | MVM-T | 0.0005 | 500 | 3 | 0.5 | 30 | 1e-07 |
| ExprGO | BoxE | 0.0001 | 1750 | 3 | 0.3 | - | - |
| ExprGO | C-HMCNN | 0.001 | 500 | 2 | 0.5 | 100 | - |
| ExprGO | MBM | 0.0001 | 1000 | 2 | 0 | - | - |
| ExprGO | MBM-T | 0.0001 | 1000 | 2 | 0 | 5 | 1e-07 |
| ExprGO | MHM | 0.0001 | 1750 | 3 | 0 | - | - |
| ExprGO | MHM-T | 0.0001 | 1750 | 3 | 0 | 30 | 1e-07 |
| ExprGO | MVM | 0.0001 | 1750 | 3 | 0.1 | - | - |
| ExprGO | MVM-T | 0.0001 | 1750 | 3 | 0.1 | 30 | 1e-07 |
| Imclef07a | BoxE | 0.001 | 1000 | 3 | 0.1 | - | - |
| imclef07a | C-HMCNN | 0.001 | 1000 | 2 | 0 | 100 | - |

**Table 4 continued from previous page**

| Dataset | Model | learning rate | hidden dim | layers | dropout | % label sample | $\nu$ |
|---|---|---|---|---|---|---|---|
| Imclef07a | MBM | 0.0001 | 1750 | 3 | 0 | - | - |
| Imclef07a | MBM-T | 0.0005 | 1750 | 3 | 0.25 | 30 | 1e-07 |
| Imclef07a | MHM | 0.001 | 1000 | 2 | 0 | - | - |
| Imclef07a | MHM-T | 0.001 | 1000 | 2 | 0 | 30 | 1e-07 |
| Imclef07a | MVM | 0.001 | 1000 | 3 | 0 | - | - |
| Imclef07a | MVM-T | 0.001 | 1000 | 3 | 0 | 30 | 1e-07 |
| Imclef07d | BoxE | 0.001 | 1750 | 2 | 0.3 | - | - |
| Imclef07d | C-HMCNN | 0.001 | 500 | 2 | 0.1 | 100 | - |
| Imclef07d | MBM | 0.0005 | 1750 | 3 | 0 | - | - |
| Imclef07d | MBM-T | 0.0005 | 1750 | 3 | 0 | 30 | 1e-09 |
| Imclef07d | MHM | 0.001 | 1750 | 2 | 0 | - | - |
| Imclef07d | MHM-T | 0.001 | 1750 | 2 | 0 | 30 | 1e-07 |
| Imclef07d | MVM | 0.001 | 1000 | 2 | 0 | - | - |
| Imclef07d | MVM-T | 0.001 | 1000 | 2 | 0 | 30 | 1e-07 |
| SpoFUN | BoxE | 0.001 | 500 | 3 | 0.3 | - | - |
| SpoFUN | C-HMCNN | 0.001 | 500 | 3 | 0.5 | 100 | - |
| SpoFUN | MBM | 0.0001 | 1000 | 3 | 0 | - | - |
| SpoFUN | MBM-T | 0.0005 | 500 | 3 | 0 | 30 | 1e-09 |
| SpoFUN | MHM | 0.0001 | 1000 | 3 | 0 | - | - |
| SpoFUN | MHM-T | 0.0001 | 1000 | 3 | 0 | 30 | 1e-07 |
| SpoFUN | MVM | 0.001 | 1750 | 3 | 0 | - | - |
| SpoFUN | MVM-T | 0.001 | 1750 | 3 | 0 | 30 | 1e-07 |
| SpoGO | BoxE | 0.001 | 500 | 3 | 0.3 | - | - |
| SpoGO | C-HMCNN | 0.001 | 1000 | 3 | 0.3 | 100 | - |
| SpoGO | MBM | 0.0001 | 1000 | 3 | 0 | - | - |
| SpoGO | MBM-T | 0.0001 | 1000 | 3 | 0 | 10 | 1e-07 |
| SpoGO | MHM | 0.001 | 500 | 3 | 0 | - | - |
| SpoGO | MHM-T | 0.001 | 500 | 3 | 0 | 30 | 1e-07 |
| SpoGO | MVM | 0.001 | 500 | 2 | 0.5 | - | - |
| SpoGO | MVM-T | 0.001 | 500 | 2 | 0.5 | 30 | 1e-07 |

### A.1 FURTHER DETAILS FOR THE BASELINES

**BoxE**: Let $p \in \mathbb{R}^d$ be a point and $z, Z \in \mathbb{R}^d$ with $z_i < Z_i$ be the lower left and upper right coordinates of a non-probabilistic label box. Then a non-probabilistic compatibility score between the point and the box can be defined as shown in Abboud et al. (2020). Concretely, Abboud et al. (2020) define per-dimension distance as

$$\text{dist}(p_i, z_i, Z_i) := \begin{cases} \frac{|p_i - c_i|}{w_i + 1}, & \text{if } z_i \leq p_i \leq Z_i \\ \frac{|p_i - c_i|}{w_i + 1} - \kappa, & \text{otherwise,} \end{cases} \tag{7}$$

where $c_i = (Z_i - z_i)/2$, $w_i = Z_i - z_i$ are the center and width of the projection of the box (i.e., an interval) in dimension $i$, and $\kappa = 0.5 * w_i * (w_i + 1 - \frac{1}{w_i+1})$ is the width dependent factor. The score function that measures the compatibility of the point $p$ and box $(z, Z)$ is defined as $s_{\text{BoxE}}(p, z, Z) = -\|v\|_2$, where $v_i = \text{dist}(p_i, z_i, Z_i)$. Given the trainable parameters $\psi^-, \psi^+ \in \mathbb{R}^{L \times d}$ for representing the labels, the lower left and the upper right coordinates of a non-probabilistic box for label $l_i$ are taken to be $z_i = \psi^-_{i,*}$ and $Z_i = \psi^-_{i,*} + \log(1 + \exp \psi^+_{i,*})$, respectively. The input $x$ is encoded as a point $p \in \mathbb{R}^d$ using a multi-layer perceptron $\mathcal{F}_\theta$, just like MVM. However, since the score is negative distance, the parameters are learnt using margin based loss given as:

$$L_{\text{margin}}(\psi, \theta; D) = \sum_{i=1}^{D} \sum_{p \in \mathcal{P}_i, n \in \mathcal{N}_i} \frac{\max\left(0, 1 - (s_{\text{BoxE}}(\mathcal{F}_\theta(x), z_p, Z_p) - s_{\text{BoxE}}(\mathcal{F}_\theta(x), z_n, Z_n))\right)}{|\mathcal{P}_i| + |\mathcal{N}_i|},$$

where $\mathcal{P}_i$ and $\mathcal{N}_i$ are the positive and negative labels for $i$-th data instance, respectively.

## A.2 COMPUTATIONAL COMPLEXITY

The computational complexity of the MBM can be divided into two parts–the computation of the instance box $\mathrm{Box}_\theta(X)$, and the computation of probability score $P(Y \mid X)$ given the encoding of the instance and label box parameters. Computing $\mathrm{Box}_\theta(X)$ amounts to splitting the output of a feed-forward network $\phi(x)$ into min and max parameters $\phi(x)^-, \phi(x)^+$, and thus has the same complexity as that of the MVM. Assuming that the operations $\sum, \prod, \log$ and $\exp$ have unit complexity, computing $P(Y \mid X)$ also has equivalent complexity. The computation of conditional probability for a single label $y_k$ using the MVM model involves computing $\sigma(\phi(x) \cdot y_k)$, an $\mathcal{O}(dL)$ operation. Now, with same assumptions for the MBM, computing $\lambda(\mathrm{Box}_\theta(x) \cap \mathrm{Box}_\psi(y_k))$ involves $d$ invocations of $\log(\exp(\cdot) + \exp(\cdot))$ followed by $d$ subtractions and $d$ invocations of $\log(1 + \exp(.))$, resulting in $\mathcal{O}(d)$ operations. The computation of $\lambda(\mathrm{Box}_\theta(x))$ omits the intersection calculation, but otherwise is the same, and thus is also $\mathcal{O}(d)$. Hence, calculating $\frac{\lambda(\mathrm{Box}(x) \cap \mathrm{Box}(y_k))}{\lambda(\mathrm{Box}(x))}$ is $\mathcal{O}(d)$, resulting in overall complexity of $\mathcal{O}(dL)$, which is equal to that of MVM. We note that the complexity of our model is not dependent on the depth of the hierarchy, as in C-HMCNN, and thus can reasonably scale to arbitrarily deep hierarchies as might be present in extreme multi-label classification. These theoretical statements are supported by Table 5, which provides average epoch duration in seconds for MVM, MBM, MBM-T, and C-HMCNN with the same hidden size. Hence, we can conclude that the computational complexity of MVM $\approx$ MBM $<$ MBM-T.

Table 5: Average epoch duration in seconds for MVM, MBM, MBM-T and CHMCNN with the same hidden size

| Model | ExprFUN | CellcycleFUN | SpoFUN | DerisiFUN |
|---|---|---|---|---|
| MVM | 17.2 | 15.4 | 15.8 | 15.5 |
| MBM | 18.4 | 16.9 | 16.3 | 15.2 |
| MBM-T | 20.1 | 26.7 | 24.1 | 27.1 |
| C-HMCNN | 17.8 | 17.0 | 15.9 | 15.4 |
| BoxE | 18.9 | 16.6 | 15.9 | 16.1 |

**Computational resources used:** For datasets with number of labels less than 500, i.e., the 4 FUN-CAT datasets, Imclef07a, Imclef07d, Diatoms and Enron, all the models were trained on TitanX GPU (memory=12GB). For the 4 GO datasets that have number of labels greater than 4000, all the models are trained on M40 GPU (memory=24GB).

## A.3 CODE

Executable python code with detailed instructions to reproduce the results reported in 1 is provided using at `https://github.com/iesl/box-mlc-iclr-2022`. We also include instructions for obtaining the pre-processed datasets, training a new model from scratch (MBM or any baseline), evaluating a pre-trained model on test set, directly downloading the pre-trained models for datasets.

## B DATASETS

The 12 datasets used in this work wary greatly in terms of domain, number of labels, number of instances, and the connectivity of label taxonomy. The characteristics of each dataset w.r.t these properties is summarized in 6. These datasets do not require a licence and are available for public usage. The links to the sources for all the datasets are provided in Table 7 below. The continuous features are standardized by removing mean and scaling to unit variance, and the categorical features are encoded as one-hot vectors.

Table 6: Summary of the datasets used in experiments. The feature based multi-label datasets span across 3 domains: functional genomics, image and text.

| Dataset | Domain | Input/Feature Type | Label Taxonomy | #Labels | #Instances | | |
|---|---|---|---|---|---|---|---|
| | | | | | Train | Val | Test |
| Expr FUN | Genomics | Continuous | Forest | 500 | 1636 | 849 | 1288 |
| Cellcycle FUN | Genomics | Continuous | Forest | 500 | 1628 | 848 | 1281 |
| Derisi FUN | Genomics | Continuous | Forest | 500 | 1608 | 842 | 1275 |
| Spo FUN | Genomics | Continuous | Forest | 500 | 1600 | 837 | 1266 |
| Expr GO | Genomics | Continuous | DAG | 4132 | 1636 | 849 | 1288 |
| Cellcycle GO | Genomics | Continuous | DAG | 4126 | 1625 | 848 | 1278 |
| Derisi GO | Genomics | Continuous | DAG | 4120 | 1605 | 842 | 1272 |
| Spo GO | Genomics | Continuous | DAG | 4120 | 1597 | 837 | 1263 |
| Diatoms | Image | Continuous | Tree | 399 | 1500 | 565 | 1054 |
| Imclef07a | Image | Continuous | Tree | 97 | 7000 | 3000 | 1006 |
| Imclef07d | Image | Continuous | Tree | 47 | 7000 | 3000 | 1006 |
| Enron | Text | Binary | Tree | 57 | 650 | 338 | 600 |

Table 7: The table provides the links to download the data from original source.

| Dataset(s) | Download Links |
|---|---|
| Imclef07a, Imclef07d, Enron, Diatoms | `http://kt.ijs.si/DragiKocev/PhD/resources/doku.php?id=hmc_classification` |
| Expr, Spo, Derisi, Cellcycle (FUN/GO) | `https://dtai.cs.kuleuven.be/clus/hmcdatasets/` |

## C SIGNIFICANCE TESTING

In section 6, we provide the results for Friedman test followed by post-hoc Nemenyi test, and also the result of pairwise Wilcoxon test where the former are not sufficient, i.e. for the case of MBM vs C-HMCNN w.r.t MAP. For the sake of completeness, table 8 provides results of pairwise Wilcoxon test comparing all models with MBM and MBM-T w.r.t all three metrics.

Table 8: The table presents the results of the Wilcoxon signed-rank test. Each cell shows the p-value for the null hypothesis that two related paired samples, here MBM/MBM-T and the other model (column), come from the same distribution, with the alternative hypothesis that MBM/MBM-T models are better in performance compared to the other model.

| Metric | | | BoxE | MVM | MHM | MBM | MVM-T | MHM-T | MBM-T | C-HMCNN |
|---|---|---|---|---|---|---|---|---|---|---|
| MAP | MBM | | 0.0005 | 0.0002 | 0.0002 | - | 0.0002 | 0.0002 | 0.4392 | 0.3386 |
| | MBM-T | | 0.0171 | 0.0261 | 0.0002 | 0.5608 | 0.0212 | 0.0002 | - | 0.4849 |
| CMAP | MBM | | 0.0007 | 0.0002 | 0.0002 | - | 0.0002 | 0.0002 | 0.3611 | 0.1506 |
| | MBM-T | | 0.0212 | 0.0134 | 0.0002 | 0.6389 | 0.0134 | 0.0002 | - | 0.3667 |
| CV | MBM | | 0.0002 | 0.0012 | 0.7407 | - | 0.0012 | 0.6890 | 1.0 | 1.0 |
| | MBM-T | | 0.0002 | 0.0002 | 0.0046 | 0.0002 | 0.0002 | 0.0046 | - | 1.0 |

## D COMPARISON TO C-HMCNN AND HMCN IN TERMS OF AU(PRC)

Due to our focus on coherence and predictive performance at the same time, we use mean average precision as the metric to evaluate predictive performance. One could, however, also use AU(PRC) as the metric for predictive performance. For the sake of completeness, in Table 9, we present the AU(PRC) values for the final models obtained in our experiments, and compare it with those provided in Giunchiglia & Lukasiewicz (2020) and Wehrmann et al. (2018a). We believe that the difference between the AU(PRC) reported for C-HMCNN in Giunchiglia & Lukasiewicz (2020) and our work is due to the difference in the hyper-parameter search strategy. Specifically, we use MAP as the reference metric to identify the best hyper-parameters while Giunchiglia & Lukasiewicz (2020) uses AU(PRC) itself.

Table 9: The table presents the area under the precision-recall curve (AU(PRC)) for the models presented in this paper and HMCN-F (Wehrmann et al., 2018a). Here, the columns with * are taken from their respective papers, and C-HMCNN is our implementation of the corresponding model for which the best hyper-parameters are also obtained, like other models in our implementation, using MAP on validation set.

| Dataset | C-HMCNN* | HCMN-F* | C-HMCNN | BoxE | MVM | MHM | MBM |
|---------|----------|---------|---------|------|-----|-----|-----|
| ExprFUN | 30.2 | 30.1 | 28.17 | 16.23 | 28.5 | 19.6 | 25.62 |
| CellcycleFUN | 25.5 | 25.2 | 24.46 | 12.25 | 21.19 | 18.19 | 21.57 |
| DerisiFUN | 19.5 | 19.3 | 17.61 | 10.19 | 14.45 | 17.42 | 18.74 |
| SpoFUN | 21.5 | 21.1 | 20.72 | 13.69 | 16.19 | 17.16 | 19.57 |
| ExprGO | 44.7 | 45.2 | 41.92 | 21.74 | 40.5 | 28.91 | 40.89 |
| CellcycleGO | 41.3 | 40 | 40.25 | 25.86 | 39.39 | 33.77 | 38.19 |
| DerisiGO | 37 | 36.9 | 36.35 | 16.22 | 34.63 | 34.59 | 36.42 |
| SpoGO | 38.2 | 37.6 | 37.09 | 23.4 | 36.54 | 34.37 | 36.38 |
| Enron | 75.6 | 72.4 | 75.18 | 72.67 | 69 | 24.76 | 74.53 |
| Diatoms | 75.8 | 53 | 70.39 | 1.86 | 68.3 | 38.29 | 78.21 |
| Imclef07a | 95.6 | 95 | 92.81 | 5.43 | 80.57 | 58.06 | 94.61 |
| Imclef07d | 92.7 | 92 | 89.29 | 7.96 | 89.73 | 64.33 | 90.48 |

# E   MAP AND CMAP

In MCL problems, where predictions consistent with a taxonomy are necessary, the model designer might be ready to sacrifice some absolute predictive performance for increased coherence, where the predictive performance can be measure using Mean Average Precision (MAP). First recall that Average Precision (AP) for an instance is the weighted mean of precisions achieved at each threshold, with the increase in recall from the previous threshold used as the weight, and Mean Average Precision (MAP) is the mean of AP across instances. Given data $D$, let $s_i^{(k)} = P(y_i = 1|x^{(k)})$ denote the score generated by the model for label $i$ given the input $x^{(k)}$. Then MAP is computed as

$$\mathrm{MAP}(s) = \frac{1}{|D|} \sum_{k=1}^{|D|} \mathrm{AP}\left(s^{(k)}, y^{(k)}\right).$$

As described in Section 6.1, given the scores $s \in \mathbb{R}^L$ produced by a model, for an input, the two post-hoc modification functions that can be applied to make these score coherent are given as

$$\delta_G^m(s)_i = \min_{l_j \in \mathrm{Anc}_G(l_i) \cup \{l_i\}} s_j, \qquad \delta_G^M(s)_i = \max_{l_j \in \mathrm{Des}_G(l_i) \cup \{l_i\}} s_j, \qquad (8)$$

where $\mathrm{Anc}_G(l)$ and $\mathrm{Des}_G(l)$ is the set of ancestors and descendants, respectively, of $l$ in the graph $G$.

One can compute MAP after applying either one of the two modification functions. If a human were to make consistent score corrections based on the complete hierarchy, they would either pick the score of a label to be the minimum of its ancestors or maximum of its descendants. Once picked, the same approach has to be applied to all the labels. We argue that in terms of correlation with human judgment, both these methods are equivalent. Hence, one can pick the modification function based on empirical performance of the post-hoc correction algorithm. We observe that $\delta_G^m$, i.e., "min of ancestors" approach, produced higher MAP with both MVM and MBM models compared to $\delta_G^M$. as can be seen in table 10. We believe that "min of ancestors" works better because it promotes sparsity in predictions, which is essential for the task of MLC.

Table 10: CMAP represents post-hoc correction of scores by taking the "minimum of ancestors", CMAP' represents post-hoc correction of scores by taking "maximum of descendants". As seen, CMAP produces better results for both MBM and MVM compared to CMAP'.

| Dataset | MVM | | | MBM | | |
| | CMAP | CMAP' | MAP | CMAP | CMAP' | MAP |
|---|---|---|---|---|---|---|
| ExprFUN | 37.66 | 14.83 | 38.37 | 38.72 | 1.78 | 38.45 |
| CellcycleFUN | 31.34 | 10.62 | 31.68 | 34.39 | 1.79 | 34.2 |
| DerisiFUN | 24.13 | 6.28 | 23.7 | 28.86 | 1.79 | 28.69 |
| SpoFUN | 25.38 | 7.4 | 25.18 | 30.27 | 1.79 | 30.1 |
| ExprGO | 41.78 | 26.15 | 44.92 | 48.56 | 0.8 | 48.45 |
| CellcycleGO | 41.09 | 21.64 | 44.25 | 44.23 | 0.8 | 44.28 |
| DerisiGO | 38.21 | 19.68 | 41.22 | 42.14 | 0.8 | 42.03 |
| SpoGO | 38.96 | 19.79 | 42.19 | 42.31 | 0.8 | 42.22 |
| Enron | 66.87 | 55.48 | 73.68 | 79.94 | 9.96 | 79.95 |
| Diatoms | 72.44 | 58.36 | 73.06 | 79.3 | 0.49 | 79.14 |
| Imclef07a | 76.56 | 69.66 | 77.14 | 91.73 | 3.12 | 91.45 |
| Imclef07d | 86.89 | 81.32 | 88.49 | 89.99 | 6.52 | 89.49 |

## F    Performance on long tail of labels

Since MVM and MBM both are embedding based models, we expect the performance of the MBM model to wary with the label frequency in the same manner as MVM. This hypothesis is corroborated by the following table which shows the spearman rank correlation between the label frequency and the mean average precision (MAP) for that label. The high positive correlation for all the models suggests that the performance of both MVM and MBM often degrades as the frequency of the labels drop. However, as shown by the table 11, both the models perform reasonably well for extremely low frequency labels. Here, tail_MAP is the mean average precision for labels with frequency 0.001% or lower.

Table 11: Comparing the performance of long tails of labels for MBM, MBM-T and MVM. Here, correlation is the spearman rank correlation between label frequency and MAP of the label and tail_MAP is the MAP for labels with frequency 0.001% or lower.

| Dataset | MBM | | MBM-T | | MVM | |
| | Correlation | tail_MAP | Correlation | tail_MAP | Correlation | tail_MAP |
|---|---|---|---|---|---|---|
| ExprFUN | 0.72 | 48.91 | 0.73 | 48.68 | 0.75 | 49.45 |
| CellcycleFUN | 0.74 | 46.11 | 0.75 | 46.17 | 0.82 | 44.02 |
| DerisiFUN | 0.79 | 41.86 | 0.82 | 41.62 | 0.79 | 41.78 |
| SpoFUN | 0.8 | 42.67 | 0.77 | 42.75 | 0.79 | 42.15 |
| Enron | 0.84 | 79.19 | 0.83 | 74.97 | 0.91 | 67.63 |
| Diatoms | 0.29 | 86.3 | 0.29 | 85.46 | 0.28 | 82.2 |
| Imclef07a | 0.73 | 59.59 | 0.74 | 60.2 | 0.82 | 58.78 |

## G    Effect of embedding dimension

To study the effect of embedding dimensions on the performance, we plot the MAP and CV vs the hidden dimension size for a subset of the datasets. As shown in figure 4, at extremely low dimensions, the MBM model consistently outperforms the the most competitive embedding based model MVM, demonstrating the usefulness of the favorable inductive bias of the box space in inducing parsimonious representations.

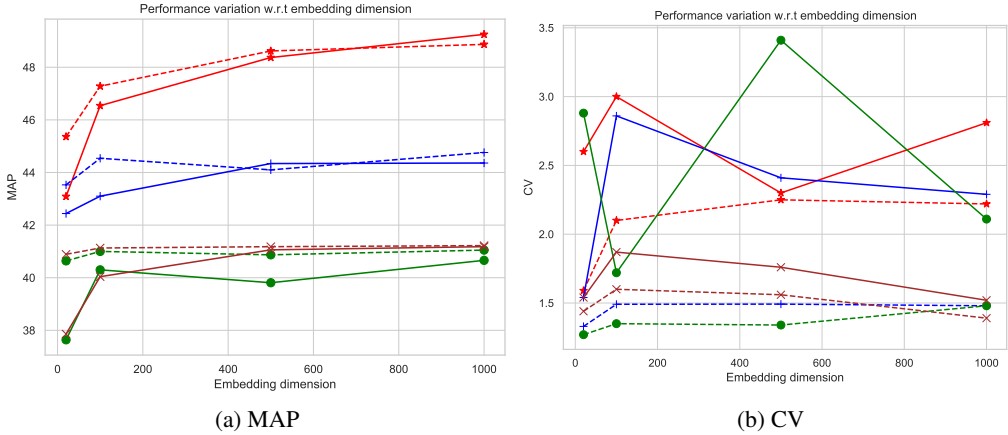

(a) MAP            (b) CV

Figure 4: Above figure shows the variation of performance w.r.t the embedding dimensions for model MBM and MVM on FUNCAT datasets. Note that the training and evaluation performed for this analysis includes the root node.

# H    EFFECT OF HIERARCHY ON THE MARGINAL SCORE

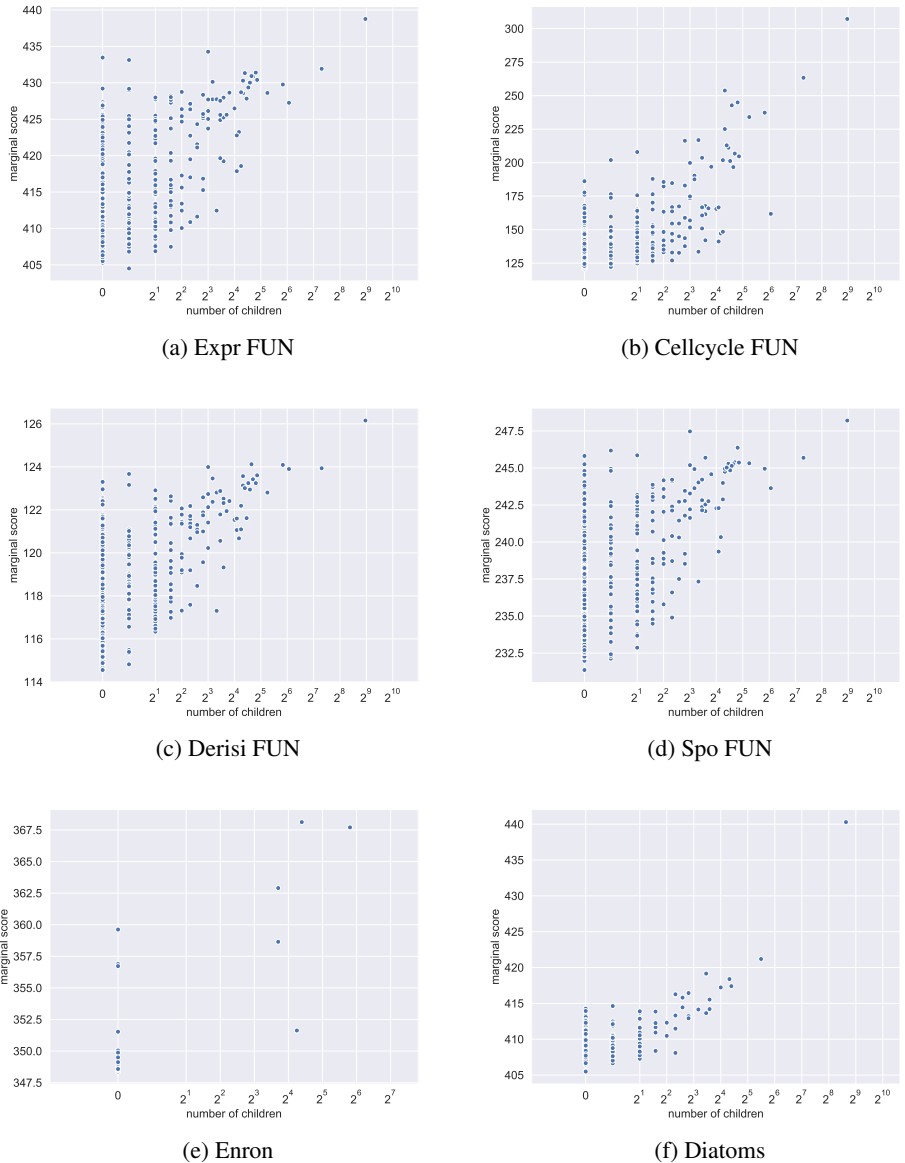

(a) Expr FUN

(b) Cellcycle FUN

(c) Derisi FUN

(d) Spo FUN

(e) Enron

(f) Diatoms

Figure 5: As discussed in Section 7, the scatter plots further show that the magnitude of box embeddings(box volume) relate to the level of generality in taxonomy.In the figure, the marginal score refers to box volume of a label and the number of children in the label space taxonomy refers to the level of generality.Note that the training and evaluation performed for this analysis includes the root node.

