# OpenReview forum: "Modeling Label Space Interactions in Multi-label Classification using Box Embeddings"
_ICLR.cc/2022/Conference — ICLR 2022 Poster_

### Official Review · Reviewer_X5cP · 2021-10-19

**Correctness:** 4
**Technical Novelty And Significance:** 3
**Empirical Novelty And Significance:** 3
**Recommendation:** 6
**Confidence:** 4

**Main Review:**

I really like the idea at the base of this paper, however I think some revisions are needed.

1. The authors should change the term “consistent” with “coherent” throughout the paper. Indeed, the authors use the word “consistent” to express the fact that the model is coherent with the taxonomy. This however, has nothing to do with the Fisher consistency and can mislead the readers.

2. At page 4 the authors first write $Y \in S$ and then $Y \in \mathcal{S}$, did you mean in both cases $Y \in \mathcal{P}(S)$?

3. At page 4, why do the authors use the function $f$? The equation can easily rewritten by substituting $f$ with its value. Also, why was $\delta$ chosen to be equal to $10^{-5}$?

4. The functional genomics datasets come in two versions, the Funcat version with a tree taxonomy, and the GO version with a forest taxonomy. I see that the authors have used the GO version for expr, cellcycle, derisi and spo, but they have used the Funcat version for diatoms. It would be interesting to have both the GO and the Funcat version for all the datasets. Further, the authors have just used Imclef07a, while Imclef07d is also available. I would recommend to test the model also on Imclef07d.

5. In order to check the statistical significance of the obtained results the authors should perform the pairwise Wilcoxon test (at least to compare the performance of MVM vs MBM and MHM vs MBM).

6. C-HMCNN does not apply a post-hoc modification, as the max layer is embedded in the network itself. Notice that post-hoc means something that is applied just at inference time (and not at training time).

7. Would it be possible to also do MVM-T?

8.  In the analysis of the learned embeddings, why weren’t the ROC curves for MHM shown?

9. In the data pre-processing, what do the authors mean by “The datasets were pre-processed to remove noisy characters, fix encoding issues”?

10. Can you also report in the appendix the final hyperparameters for each model?

11. Why is there such a big difference between the performances of C-HMCNN (as implemented by the authors) and your implementation?

12. Why aren’t the results in terms of AU(PRC) of MHM reported?


Minor comments:
- It should be C-HMCNN and not CHMCNN
- Add upward arrows in Table 2 next to MAP and CMAP

**Summary Of The Paper:**

In this work the authors propose a new model for multi-label classification problems. In their model, the authors represent the labels as boxes and they use neural networks to embed the input datapoints in the same space. Thanks to the box representation, the model achieves higher interpretability, and the authors can inject background knowledge (in the form of taxonomies) into the model to improve the coherence of the learnt labels representation with the taxonomy.
The model has been tested on 7 publicly available datasets, and evaluated using three different metrics, namely, CMAP, CV and and MAP.


**Summary Of The Review:**

The paper shows an interesting idea, however to have full acceptance some revisions are needed (see full review).

---

> ### Author Response · Authors · 2021-11-23
> **Response to reviewer 4 (X5cP) [Part 1]**
>
> Thank you for the detailed and objective feedback! We believe we have addressed all of your concerns through this response and by making a revision to the submitted draft. Since the response is long, we break it down into two parts.
>
> > The authors should change the term “consistent” with “coherent” throughout the paper.
>
> This change is reflected in the most recent revision of the paper.
>
>
> > At page 4 the authors write $Y\in S$ and then $Y \in \mathcal S$
>
> $S$ is the set of all joint labelings and a labeling $Y$ is an element of this set. Hence, in both cases on page 4 we mean $Y\in S$.
>
> > At page 4, why do the authors use the function $f$? The equation can easily be rewritten by substituting $f$ with its value.
>
> Indeed, it is much clearer to write the definition without using $f$. We have updated the paper to reflect this.
>
>
> > Also, why was $\delta$ chosen to be equal to $10^{−5}$?
>
> We want to represent the input $x$ as boxes of fixed but small width. We use  $\delta=10^{-5}$ because it is numerically stable and gives good performance. However, one could use slightly different values of $\delta$ and obtain very similar performance.
>
> > I see that the authors have used the GO version for expr, cellcycle, derisi and spo, but they have used the Funcat version for diatoms. It would be interesting to have both the GO and the Funcat version for all the datasets. Further, the authors have just used Imclef07a, while Imclef07d is also available. I would recommend to test the model also on Imclef07d.
>
> To clarify, in the initial draft submitted, we presented results on the FunCat versions of Expr, Cellcycle, Derisi and Spo. However, based on your suggestion, we performed evaluation on GO versions of these datasets as well as Imclef07d, and included the results in the latest revision. We could only find one version of [Diatoms](http://kt.ijs.si/DragiKocev/PhD/resources/doku.php?id=hmc_classification#diatoms) dataset that we had already included in our initial draft. To conclude, we now have included evaluation results on 12 datasets.
>
> > In order to check the statistical significance of the obtained results the authors should perform the pairwise Wilcoxon test (at least to compare the performance of MVM vs MBM and MHM vs MBM)
>
> We have updated the paper to include the critical diagrams for the Nemenyi test and we also provide the results of the pairwise Wilcoxon test comparing MVM and MBM. The results are as follows:
> 1. In terms of predictive performance (MAP), MBM is significantly better than all other models except MVM, to which it is comparable (p-value for Wilcoxon test is 0.1).
> 2. In terms of coherence (CV and CMAP), MBM performs significantly better than MVM.
>
> > C-HMCNN does not apply a post-hoc modification, as the max layer is embedded in the network itself. Notice that post-hoc means something that is applied just at inference time (and not at training time).
>
> Thank you for the clarification regarding the technical use of “post-hoc”. We were aware that C-HMCNN incorporates the max layer into the network itself, however this may not have been conveyed accurately in our original draft. We have now modified Section 6.1 to provide a more accurate representation of C-HMCNN.

---

> > ### Author Response · Authors · 2021-11-23
> > **Response to reviewer 4 (X5cP) [Part 2]**
> >
> > In this second part for the two-part response, we address the points not covered in part one.
> >
> > > Would it be possible to also do MVM-T?
> >
> > We tried to formulate MVM-T by using dot product between the label embeddings as the compatibility score, however this did not perform well. Instead we used a bilinear map (a common choice for representing graphs with vectors) to create an MVM-T baseline. Please see the updated section 5 for the exact details. Note that the bilinear scoring function is only used in MVM-T--the MVM model remains the same. We have included the results for MVM-T  in the paper. We find that MVM-T does not improve w.r.t any metric when compared to MVM.
> >
> > > In the analysis of the learned embeddings, why weren’t the ROC curves for MHM shown?
> >
> > The ROC curves for MHM were omitted because the original figure was already very cluttered. We now have a cleaner figure that also includes the ROC curves for MHM. Refer to the updated figure 3.
> >
> > >In the data pre-processing, what do the authors mean by “The datasets were pre-processed to remove noisy characters, fix encoding issues”?
> >
> > Thanks for catching this mistake. Initially, we were also pre-processing some candidate datasets that contained raw text. However, we did not go through with those experiments but forgot to omit this line. We have removed this line in the latest revision.
> >
> > > Can you also report in the appendix the final hyperparameters for each model?
> >
> > The final hyper-parameters have been included in Appendix A.
> >
> > > Why is there such a big difference between the performances of C-HMCNN (as implemented by the authors) and your implementation?
> >
> > We followed the description of the model provided in the paper to generate our implementation. However, due to this question, we went to the implementation provided by the authors. While combing through the code, we found that their implementation removes certain labels from evaluation (1 label for all FUNCAT, Diatoms, Imclef datasets and 4 labels for the GO datasets). Since their paper does not mention this pre-processing step anywhere, we did not incorporate it in our evaluation. We believe this might be the main reason for the reported difference in the performance of the two implementations.
> >
> > > Why aren’t the results in terms of AU(PRC) of MHM reported?
> >
> > We have added a column with the results of MHM in terms of AU(PRC) in Table 9.

---

> > > ### Comment · Reviewer_X5cP · 2021-11-23
> > > **not removed the root nodes of the hierarchies in the dataset?**
> > >
> > > Dear authors,
> > >
> > > thank you very much for your responses, I see that the paper has greatly improved from the first version.
> > >
> > > I have though a major concern regarding your experimental analysis.
> > >
> > > As outlined in https://dtai.cs.kuleuven.be/clus/hmcdatasets/ (where you can find all the datasets for HMC problems) the root of the hierarchies should always be removed. This is due to the fact that they are just dummy labels used to complete the hierarchy. If I remember correctly, they are always equal to 1, and this might really boost your performances.
> > >
> > > If that is the case, the experimental analysis should be re-run and the new results should be reported.

---

> > > > ### Author Response · Authors · 2021-11-24
> > > > **We are re-running the experiments**
> > > >
> > > > We believe that the trend of the results will hold even when the root nodes are removed. However, we acknowledge your concern and have started re-running the experiments. We will try to provide the updated results as soon as possible.

---

### Official Review · Reviewer_HdXC · 2021-10-30

**Correctness:** 3
**Technical Novelty And Significance:** 3
**Empirical Novelty And Significance:** 2
**Recommendation:** 5
**Confidence:** 3

**Main Review:**

Strengths:

The idea to use box embeddings for hierarchical multi-label classification seems good and promising.

The paper has mostly been written clearly.

Weaknesses:

The proposed method is a relatively straight-forward application of box embeddings as developed in earlier literature (including e.g. Dasgupta et al., 2020a) into hierarchical multi-label classification.

I am not sure I understand why one would need 'a model that uses box embedding to capture general label-label relationships without the explicit use of label taxonomy'. Would it not be beneficial to explicitly use the label taxonomy? The paper writes that requiring a 'complete label taxonomy either at inference time or both at training as well as inference time, making these models hard to scale to large label spaces'. I would like to see concrete examples of when this really happens. To me it seems that it is relatively easy to learn a mostly complete label taxonomy as the first task and then use e.g. CHMCNN.

No clear enough definition has been provided for the MAP measure. Is the mean taken over the labels and for each label there is an area under the precision-recall curve calculated?

It has not been explained sufficiently why MAP is a good measure to look at. As the paper correctly states, 'it does not take into account inconsistencies in the predicted scores w.r.t. the label taxonomy'. Instead of MAP, one could consider different score thresholds and for each threshold calculate how many labels would get correct predictions. However, indeed, Giunchiglia & Lukasiewicz, 2020 also seems to use the same MAP measure.

The baselines in the experiments could have included HMCN-R and HMCN-F from Giunchiglia & Lukasiewicz, 2020. While it is true that 'HMCN does not try to enforce consistency strongly and focuses solely on predictive performance', it does not mean that HMCN could not be a strong competitor in the experiments.

The origin of the MVM (Multi-label Vector Model) as a baseline has not been explained or referenced. What is the motivation behind this method?

Statistical significance of the differences of MBM over MVM has not been demonstrated through a statistical test. Currently the text about Table 2 states that 'We observe that the predictive performance of MBM measured using MAP is better than that of MVM and MHM on 5 out of the 7 datasets.'

It is not clear whether the label interaction loss has been used as a regularizer on top of the main NLL-loss or in some other way. If as regularizer, then is there a regularization parameter lambda defining how much each loss contributes to the overall loss? How is this parameter lambda tuned?

Supplementary figure 3 shows the trajectories of MAP, but what would be the trajectories of CV and CMAP?

The interpretability claims seem too bold in my opinion. It is still hard to interpret high-dimensional boxes and their overlaps. The experiments would benefit from more baselines.

page 9: 'hyperberbolic' -> 'hyperbolic'


**Summary Of The Paper:**

The paper proposes a method using box embeddings to perform hierarchical multi-label classification.


**Summary Of The Review:**

The paper is interesting and results seem to be quite good, but the justification of why availability of the label taxonomy should not be available is a bit lacking. The proposed method is quite a simple step from existing literature, and there are several shortcomings about the text highlighted above. The experiments would benefit form more baselines.

---

> ### Author Response · Authors · 2021-11-14
> **Regarding inclusion of HMCN as a baseline in Table 2.**
>
> Thank you for the insightful comments! We will be addressing all your concerns in a future response. However, before that we wanted to get some more details about the comments on the baselines made in the following lines:
>
> > The baselines in the experiments could have included HMCN-R and HMCN-F
>
> In the submitted draft, **we do compare with HMCN. These results are presented in Table 7 in Appendix E**, but are omitted from Table 2 due to the following two reasons:
>
> 1. Since both C-HMCNN and HMCN are designed to work with explicit label taxonomy provided during training and inference, these methods are not directly comparable to the central method proposed in our paper, i.e., MBM. However, for the sake of comparison with the state-of-the-art for hierarchical MLC, we do include C-HMCNN in Table 2 (Giunchiglia & Lukasiewicz (2020) show that C-HMCNN is indeed the state-of-the-art). Since we have other interesting baselines like MVM and MHM that are directly comparable to MBM, we find limited value in having HMCN also in the already crowed Table 2.
>
> 2. Another reason to move HMCN to the appendix was the unavailability of their code and a mismatch in the metric. The metric AU(PRC) used in Wehrmann et. al. performs a global mean of average precision, i.e., simultaneous average across labels as well as instances, while MAP, which is the metric in our analysis, performs mean of average precision across instances. As noted in Giunchiglia & Lukasiewicz (2020), the code for HMCN is not available for us to add MAP, CMAP and CV metrics in their code and re-run the experiments. So instead, we updated our code to also report AU(PRC) along with MAP. However, adding a column for HMCN in Table 2 resulted in 21 cells--the ones corresponding to MAP, CMAP, and CV--being empty. Hence, we considered moving the HMCN results to the appendix as the best choice.
>
> Regarding the following comment:
>
> > The experiments would benefit from more baselines
>
> From the current state of the literature, we were only able to identify hyperbolic embeddings and Euclidean vector embeddings as two methods that use label embeddings for "Modeling Label Space Interactions in Multi-label Classification", which is the central idea behind this paper. Hence, we incorporate these two methods in our baselines. Moreover, we also compare with the general state-of-the-art method in hierarchical multi-label classification, i.e., C-HMCNN. We feel that the choice of baselines in our paper provide sufficient support for the claims made within. **That being said, we would like to know if you have a specific reason for adding another baseline and a candidate baseline for it?**

---

> ### Author Response · Authors · 2021-11-23
> **Full response to reviewer 3 (HdXC) [Part 1]**
>
> Thank you for the feedback! We try to address all your concerns and questions below.
>
> > The proposed method is a relatively straight-forward application of box embeddings as developed in earlier literature (including e.g. Dasgupta et al., 2020a) into hierarchical multi-label classification.
>
> The use of box embeddings has been restricted to the problems of graph representation and KB completion where the node or entities are static elements. In the case of MLC, the input $x$ is not an entity and has to be encoded as a box in the same space as the label boxes. There are several ways to do this, however, not all of them perform well. For instance, look at the performance of the newly added baseline BoxE and the discussion with reviewer 1.
> Moreover, we also present extensive analysis on the coherence of the model's predictions and the properties of the learned label embeddings--all of which is novel and interesting.
>
> > I am not sure I understand why one would need 'a model that uses box embedding to capture general label-label relationships without the explicit use of label taxonomy'. Would it not be beneficial to explicitly use the label taxonomy? The paper writes that requiring a 'complete label taxonomy either at inference time or both at training as well as inference time, making these models hard to scale to large label spaces'. I would like to see concrete examples of when this really happens. To me it seems that it is relatively easy to learn a mostly complete label taxonomy as the first task and then use e.g. C-HMCNN.
>
> Indeed, in our preliminary experiments, we tried to first identify the edges of the label taxonomy from the data using the incomplete and noisy data assumption. Roughly speaking this amounts to the following: if label A and B co-occur in some instances does not mean that they are connected in the label taxonomy (noisy data assumption), and if A and B never co-occur in the given data, it does not mean that there cannot be an edge between them in the label taxonomy (incomplete world assumption). As also noted by reviewer 2, one can only get a noisy taxonomy from the data, and, as we observed in our preliminary experiments, enforcing constraints from such a taxonomy can result in low predictive performance. In such a setting, a model that can learn without the provision of explicit taxonomy is very useful.
>
> > No clear enough definition has been provided for the MAP measure. Is the mean taken over the labels and for each label there is an area under the precision-recall curve calculated?
>
> The following sentence
> > "The Mean Average Precision (MAP), that is the **mean of the average precision values across instances in the test set** is used as the evaluation metric" to describe MAP.
>
> in our initial submission indicates that the mean is taken over the instances where each instances' value is its average precision. We understand that this might not be clear enough, so in the latest revision, we also express this using an equation in Appendix E.
>
> > It has not been explained sufficiently why MAP is a good measure to look at. As the paper correctly states, 'it does not take into account inconsistencies in the predicted scores w.r.t. the label taxonomy'. Instead of MAP, one could consider different score thresholds and for each threshold calculate how many labels would get correct predictions. However, indeed, Giunchiglia & Lukasiewicz, 2020 also seems to use the same MAP measure.
>
> MAP and AU(PRC) are two standard measures of performance for general MLC. We could take various thresholds and compute the accuracy at each of these, however, we argue that MAP is a smoother and more stable version of precisely this idea. Since MAP does not take into consideration the incoherence/inconsistency in the predicted scores, we also compute constraint violation to measure the inconsistencies. Finally, we also try to combine both these aspects and compute the CMAP value for each model. To the best of our knowledge, there isn't any standard metric that takes into account both coherence and predictive performance simultaneously.
>
> > The origin of the MVM (Multi-label Vector Model) as a baseline has not been explained or referenced. What is the motivation behind this method?
>
> The MVM model is merely a special instance of multi-layer perceptron used in the context of general multi-label classification--we simply interpret the final layer as label embeddings in the form of vectors in Euclidean space. Although it does not follow constraints specified by the taxonomy, as shown by the results in this paper, as far as predictive performance goes, it is an extremely competitive baseline.

---

> > ### Author Response · Authors · 2021-11-23
> > **Full response to reviewer 3 (HdXC) [Part 2]**
> >
> > In this response, we address the concerns not covered in Part 1.
> >
> > > Statistical significance of the differences of MBM over MVM has not been demonstrated through a statistical test.
> >
> > We have added Nemenyi test's critical diagrams and the result of Wilcoxon test in Section 6 to check the statistical significance of our claims. We find that:
> > 1. In terms of predictive performance (MAP), MBM is significantly better than all other models except MVM, to which it is comparable.
> > 2. In terms of coherence (CV and CMAP), MBM performs significantly better than MVM and MHM.
> > 3. MBM performs significantly better than C-HMCNN in terms of MAP and CMAP even though MBM does not require the label taxonomy explicitly.
> >
> > > It is not clear whether the label interaction loss has been used as a regularizer on top of the main NLL-loss or in some other way. If as regularizer, then is there a regularization parameter lambda defining how much each loss contributes to the overall loss? How is this parameter lambda tuned?
> >
> >
> > We apologize for not being clear about this. Yes, the label interaction loss is added as a regularizer on MVM, MBM and MHM models to obtain MVM-T, MBM-T and MHM-T models, respectively. We have made this more explicit in section 4.2. We tune the weight of the regularizer along with other hyper-parameters using the performance on the validation set. The final hyperparameters are provided in Appendix A.
> >
> > > Supplementary figure 3 shows the trajectories of MAP, but what would be the trajectories of CV and CMAP?
> >
> > We have included the trajectories for CMAP as well as CV in Figure 4 in Appendix G in the latest revision of the paper.

---

### Official Review · Reviewer_F1ga · 2021-11-02

**Correctness:** 4
**Technical Novelty And Significance:** 3
**Empirical Novelty And Significance:** 3
**Recommendation:** 8
**Confidence:** 3

**Main Review:**

Strengths:
- Most existing multi-label classification methods predict labels that are inconsistent with natural/latent taxonomic constraints.
- This work introduces the multi-label box model (MBM), a method for multi-label classification that combines the encoding power of neural networks with the inductive bias and probabilistic semantics of box embeddings (trainable Venn-diagrams based on hyper-rectangles). This enables MBMs to capture taxonomic relations among labels.
- MBMs can be trained via gradient descent from data.
- Box embeddings can be read as calibrated conditional probabilities.
- The model is naturally more interpretable than other methods for multi-label classification.
- Partial information about label-label relationships can be injected into the model training, improving the model's consistency. This is done via the "label interaction loss".
-  MBM is theoretically grounded, and it can improve taxonomic consistency while preserving or surpassing state-of-the-art predictive performance.
- Multi-label classification is an important problem that has real world applications and is relevant to the ML/AI community. Enforcing taxonomic constraints in multi-label classification is an important subproblem.
- The approach seems quite novel.
- The method is well-framed in the existing literature.
- The method is very-well motivated in theory.
- The baseline models for comparison seem reasonable.
- Evaluations are conducted over 7 benchmark datasets, which provide label space taxonomies which are additionally useful for evaluating the learned taxonomy.
- The evaluation metrics seem reasonable.
- The evaluation results are promising.
- Experimental results suggest that the label embeddings seem to capture the latent label taxonomy without it being explicitly provided.
- the approach can reasonably scale to arbitrarily deep hierarchies as might be present in extreme multilabel classification.


Weaknesses:
- Multi-label classification is a well-studied problem, and there has been work on enforcing taxonomic constraints for multi-label classification in the past. This somewhat limits novelty; however, the approach seems novel, so this is not a major concern.
- The encoder neural network tested is relatively simple: a three-layer MLP; it would be interesting to see whether this method generalizes to more complex neural networks (i.e., can good box representations still be learned with deeper networks on more challenging problems).
- It would be nice to see which results are statistically significant in Table 2.
- Figure 2 is a bit hard to interpret due to the large number of settings.
- Evaluations w.r.t. the taxonomy quality assume the ground-truth taxonomies are perfect, but in practice, there might be a human-bias in how labels are grouped. It would be interesting to see if you could reverse engineer the taxonomy from the embeddings and see if humans agree that the taxonomy makes logical sense and doesn't have many atypical/meaningless relationships detected.

Additional Questions/Comments:
- Is there any disadvantage to using hyper-rectangles over different geometric assumptions (e.g., hyperspheres)?
- Since the method naturally outputs calibrated probabilities and utuilizes taxonomic constraints, I wonder if it is more robust to imbalance problems that tend to plague multi-label classification. Do you have any insights into whether this is true or not?
- The improvements in the metrics seem relatively small when inroducing taxonomy information in the training. Do you think this is because the base model is already capturing model of the latent taxonomic structure and doesn't need the additional guidance.

**Summary Of The Paper:**

Most existing multi-label classification methods predict labels without considering natural/latent taxonomic constraints. This work introduces the multi-label box model (MBM), a method for multi-label classification that combines the encoding power of neural networks with the inductive bias and probabilistic semantics of box embeddings (trainable Venn-diagrams based on hyper-rectangles). MBM has some nice properties: they can be trained via gradient descent from data, and box embeddings can be read as calibrated conditional probabilities. Experiments are conducted on seven benchmark datasets with promising results, and results suggest that the label embeddings seem to capture the latent label taxonomy.

**Summary Of The Review:**

Multi-label classification is an important problem that has real world applications and is relevant to the ML/AI community. Enforcing taxonomic constraints in multi-label classification is an important subproblem. The proposed approach is a novel solution to this subproblem, and is well-framed in the literature and theoretically-grounded. the paper is relatively well-structured. Experiments and analysis seem sound, and experimental results are proming: MBM is improve taxonomic consistency while preserving or surpassing state-of-the-art predictive performance on several benchmark datasets compared to reasonable baselines.

---

> ### Author Response · Authors · 2021-11-23
> **Response to reviewer 2 (F1ga): Addressing the concerns**
>
> Thank you for the positive feedback. Below, we try to address all your concerns and questions.
>
> ### Weaknesses:
>
> > Multi-label classification is a well-studied problem, and there has been work on enforcing taxonomic constraints for multi-label classification in the past. This somewhat limits novelty; however, the approach seems novel, so this is not a major concern.
>
> We appreciate the reviewer’s acknowledgment that our approach is novel. We would highlight, additionally, that our model (MBM) does not require the injection of taxonomic constraints, and while such constraints can be easily injected (MBM-T) we actually outperform the current SOTA approach C-HMCNN both with and without these constraints. A further novelty is that the probabilistic nature of our model allows for this method to be extended to other structured prediction tasks, where explicit hierarchies are not available but softer probabilistic constraints are.
>
> > The encoder neural network tested is relatively simple: a three-layer MLP; it would be interesting to see whether this method generalizes to more complex neural networks (i.e., can good box representations still be learned with deeper networks on more challenging problems).
>
> We believe that it is imperative to check MBM's usefulness thoroughly in a more controlled setting with fewer orthogonal design variables. We do this by evaluating on 12 feature-based datasets that have different characteristics and performing analysis of the learned label embeddings. Given the findings of this paper, we believe that expanding this framework to larger neural networks, possibly to pre-trained networks for image and text domains, is an exciting future direction to pursue. We note that prior work has demonstrated that deep neural networks (eg. LSTMs, transformers) are capable of training good box representations. [0,1]
>
> > It would be nice to see which results are statistically significant in Table 2.
>
> We have added Nemenyi test's critical diagrams and the result of Wilcoxon test in Section 6 to check the statistical significance of our claims. We find that:
> 1. In terms of predictive performance (MAP), MBM is significantly better than all other models except MVM, to which it is comparable.
> 2. In terms of coherence (CV and CMAP), MBM performs significantly better than MVM and MHM.
> 3. MBM performs significantly better than C-HMCNN in terms of MAP and CMAP even though MBM does not require the label taxonomy explicitly.
>
>
> > Figure 2 is a bit hard to interpret due to the large number of settings.
>
> Acknowledging the difficulty in parsing the figure, we have modified figure 2 (now figure 3 in the latest revision) to show the envelope of the ROC curves instead of the individual curves and also added a complete table with the AUC(ROC) values.
>
> > Evaluations w.r.t. the taxonomy quality assumes the ground-truth taxonomies are perfect, but in practice, there might be a human-bias in how labels are grouped. It would be interesting to see if you could reverse engineer the taxonomy from the embeddings and see if humans agree that the taxonomy makes logical sense and doesn't have many atypical/meaningless relationships detected.
>
> Indeed this would be an interesting use for the MBM. However, performing such an evaluation is extremely resource-intensive and hence could not be performed in the scope of this work.

---

> > ### Author Response · Authors · 2021-11-23
> > **Response to reviewer 2 (F1ga): Answer to additional questions.**
> >
> > Below we provide the answers to the additional questions.
> >
> > ### Additional Questions/Comments:
> >
> > > Is there any disadvantage to using hyper-rectangles over different geometric assumptions (e.g., hyperspheres)?
> >
> > The primary disadvantage of boxes is they are convex (a property shared by hyperspheres) or, more generally, simply connected. This is somewhat representationally limiting - in this case, it forces the encoder to embed the inputs into a simply-connected region, a property which is preserved under continuous transformations, and there is no prior reason to assume the input features corresponding to a particular label are, themselves, simply-connected. One option to ameliorate this issue would be to use multiple box representations for each label, however, this also increases complexity. On the other hand, this connectivity prior may actually provide a useful inductive bias for the model in settings where it actually is a reasonable assumption for the inputs.
> >
> > In comparison to using other connected or even convex representations, boxes do not appear to have a significant disadvantage, and their virtues (straightforward calculation of intersection volumes) are necessary for a probabilistic interpretation and are easily found in other representation schemes. In addition, notice that the number of *corners* for an interval, rectangle and cube is 2, 4, 8, respectively- i.e. the number of corners of a box increases exponentially with dimensions. This allows for more boxes to intersect with a given box without overlapping with each other. We believe that this is an important property that gives more expressivity to a hyper-rectangle based model compared to one based on hyper-spheres. This is an intuitive argument, and verifying this empirically could be an interesting direction of future work.
> >
> > > Since the method naturally outputs calibrated probabilities and utilizes taxonomic constraints, I wonder if it is more robust to imbalance problems that tend to plague multi-label classification. Do you have any insights into whether this is true or not?
> >
> > We did wonder about this question. Hence, we performed an analysis of the performance of MVM and MBM on the long tail of infrequent labels. However, we could not conclude that MBM provides a significant advantage over MVM on the long tail. This analysis was reported in Appendix F titled "Performance on the long tail of labels" in the originally submitted draft and is also retained in the latest revision.
> >
> > > The improvements in the metrics seem relatively small when introducing taxonomy information in the training. Do you think this is because the base model is already capturing the model of the latent taxonomic structure and doesn't need the additional guidance.
> >
> > Indeed, as suggested by the label embedding analysis the label embeddings capture the taxonomy very well even without the explicit label interaction loss.
> > However, we also find that in order to maintain the predictive performance, the influence of label interaction loss on training has to be kept on the lower side.  For instance, one cannot first train the label embeddings solely using the taxonomy, nor can one pick and fix an arbitrary configuration for label embeddings that satisfies all taxonomic constraints as both these hinder the learning of the encoder parameters $\theta$ leading to low predictive performance (MAP).
> > Hence, we find that applying the label interaction loss with a low weight almost always improves the CV (11/12 datasets) but does not improve the predictive performance (MAP) significantly.
> >
> > [0] Vilnis, Luke, Xiang Li, Shikhar Murty, and Andrew McCallum. "Probabilistic embedding of knowledge graphs with box lattice measures." arXiv preprint arXiv:1805.06627 (2018).
> >
> > [1] Onoe, Yasumasa, Michael Boratko, Andrew McCallum, and Greg Durrett. "Modeling fine-grained entity types with box embeddings." arXiv preprint arXiv:2101.00345 (2021).

---

> > > ### Comment · Reviewer_F1ga · 2021-11-28
> > > **Thank you for your response!**
> > >
> > > Thank you for the careful response to my comments. I think they satisfactorily address my concerns, so I am maintaining my rating of "accept".

---

### Official Review · Reviewer_Eo7g · 2021-11-02

**Correctness:** 4
**Technical Novelty And Significance:** 1
**Empirical Novelty And Significance:** 3
**Recommendation:** 5
**Confidence:** 4

**Main Review:**

Strengths:

- The paper tackles an interesting problem and it also applies when we have no access to the taxonomy.
- The presentation is relatively clear and self contained.
- The technical details appear sound.

Weaknesses:

- The novelty/originality is limited: box embeddings, probabilistic semantics of their intersections, gumbel boxes, bessel volume, etc are all adopted from existing literature. Propositions and the corollary appear more like observations rather than results.

- The overall setup is not well-motivated. While I see why box embeddings can be useful for encoding class interactions and their hierarchies (which are in the form of directed acyclic graphs), it is not very clear to me whether one needs a probabilistic semantics as is used in this work.

- The BoxE paper (Abboud et al, BoxE: A Box Embedding Model for Knowledge Base Completion, NeurIPS, 2020), which is also a box embedding model (proposed for link prediction), is very relevant for this work: It shows that boxes can capture arbitrary relational hierarchies in the space. This is shown for binary relations and therefore it is more general, but it clearly applies to class hierarchies which are only a special case. This suggests that one can model hierarchies using box embeddings in an even simpler way, where box containment implies a subclass relationship (without the probabilistic interpretation). If we already know the class taxonomy, we can even inject this information to the space, i.e., if C is a subclass of D, one can enforce the corresponding C-box to be contained in the corresponding D-box in the space, etc. That is, if the class taxonomy is known, one can provably enforce these using the ideas presented in (Abboud et al). If the taxonomy is not known, then it can still be learned.

- The probabilistic interpretation of boxes may bring in some value, but it is not clear to me whether this is the case, after reading the paper. In fact, the above-outlined approach would achieve the similar goals in a simpler way in my understanding - This may possibly result in an even stronger baseline than the given baselines. Notice that when a C-box is contained in the D-box after training, any prediction for class C will be consistent, since it will also be a D by the space configuration (and there are many possible configurations that can achieve the same thing).




**Summary Of The Paper:**

The problem tackled in this paper is that of multi-label classification, where class labels form a taxonomy (i.e., a hierarchy), and the goal is to enforce the predictions to comply with the label taxonomy.  This is a well-known problem in machine learning and it is important as it imposes a more structured learning space, by implicitly enforcing predictions to be consistent with the label taxonomy. The approach taken in this paper is based on box embeddings, where the idea is to view classes as boxes in the space, and apply a certain probabilistic semantics of box embeddings to eventually model taxonomic relations between classes.


**Summary Of The Review:**

The paper is well-written and interesting but limited in originality and novelty given works of Vilnis et al., Dasgupta el al. It is an interesting application of earlier findings to the domain of multi-label classification, but it is lacking in motivation for the precise model choices. The comparison with earlier works is limited without which the significance of the work is unclear, which led me to suggest a weak reject.

---

> ### Author Response · Authors · 2021-11-12
> **A clarification question for reviewer Eo7g**
>
> Thank you for the valuable feedback! We will address all points you raised more completely in future responses; however, we first wish to clarify the points raised regarding BoxE (bullets 3 and 4 in the *Weaknesses* section).
>
> Specifically, **could you provide more details as to how BoxE could be applied to the multi-label classification setting?** BoxE (as shown in Example 1 in Abboud et al.) uses two boxes $r^{(1)}$ and $r^{(2)}$ for one relation (which is a binary relation in our case), and represents entities by two vectors--a base position $e$ and translational bump $b$. The most direct translation of this model to a multi-label classification task would be to use a neural network $f$ to encode the input $x$ as a base vector and translational bump, i.e. $(e_x,b_x):=f(x)$, and represent each label $l$ through a base vector and translational bump parameterized via free parameters, i.e. $(e_l, b_l)$. In this case, the solitary binary relation, which we can call "Labeled-As",  would be encoded as two boxes. With this formulation, it is unclear how consistency/coherence to a label hierarchy would be enforced. If you could provide additional details describing the intended approach, we would greatly appreciate it!

---

> > ### Comment · Reviewer_Eo7g · 2021-11-12
> > **Clarification**
> >
> > It's easier; please note that BoxE applies to relations of any arity, which clearly includes classes. A class (or, label) $C$ is captured by a single box (not two as in a relation), say $C_B$, and in this case no bumps apply because a class takes only one entity as an argument. Then, assuming $f$ is used to encode the input $x$ as a base vector, and assuming the label of $x$ is $C$, all you need is to optimise $x$ to be close to the center of the box $C_B$, which it belongs to.

---

> > > ### Author Response · Authors · 2021-11-13
> > > **Probabilistic semantics and the limited value of adding BoxE as a baseline**
> > >
> > > Thank you for the quick response and clear explanation!
> > >
> > > If the classes are represented as unary relations in the BoxE framework, the main difference between this approach and our own model is the score function used.
> > >
> > > **With this awareness, there are a plethora of score functions which could be considered:**
> > > 1. **Query2Box distance** [Ren et al. (2020)](https://openreview.net/pdf?id=BJgr4kSFDS).
> > >
> > > 2. **Hard Box distance** [Subramanian and Chakrabarti (2018)](https://dl.acm.org/doi/pdf/10.1145/3209978.3210150).
> > >
> > > 3. **Surrogate hard box distance** [Vilnis et al. (2018)]([https://aclanthology.org/P18-1025.pdf](https://aclanthology.org/P18-1025.pdf)).
> > >
> > > 4. **Smooth Box distance** [Li et al. (2019)](https://openreview.net/forum?id=H1xSNiRcF7).
> > > 5. **Gumbel Box distance** [Dasgupta et al. (2020)](https://openreview.net/forum?id=P5-ggsKvS3K) (which we use)
> > >
> > > 6. **BoxE distance** [Abboud et al. (2020)]([https://proceedings.neurips.cc/paper/2020/hash/6dbbe6abe5f14af882ff977fc3f35501-Abstract.html](https://proceedings.neurips.cc/paper/2020/hash/6dbbe6abe5f14af882ff977fc3f35501-Abstract.html))
> > > 7. **Trained temperature box distance** [Boratko et al. (2021)](https://openreview.net/forum?id=0IqTX6FcZWv)
> > >
> > > Note: The box-box distances mentioned above work as intended even if one box is taken to be infinitesimally small, i.e., a point.
> > >
> > > BoxE distance does not seem to hold any particular benefits over the others when viewed in this context. The only motivation given for the specific formulation of BoxE distance was:
> > > > The idea is to define a function that grows slowly if a point is in the box (relative to the center of the box), but grows rapidly if the point is outside of the box, so as to drive points more effectively into their target boxes and ensure they are minimally changed and can remain there once inside.
> > >
> > > This property is possessed by all the distance functions mentioned above, in particular, the one we use in this paper, i.e., $P_\text{MBM}$.
> > >
> > > You highlighted a few additional benefits of BoxE, which our model shares and *expands* upon:
> > >
> > > 1.  **Representational capacity**: You mention that “[BoxE] shows that boxes can capture arbitrary relational hierarchies in the space”, a statement that applies equally well to our model (Proposition 3.).
> > >
> > > 2.  **Simplicity**: You suggest that “one can model hierarchies using box embeddings in an even simpler way, where box containment implies a subclass relationship (without the probabilistic interpretation)”. Although justifying the rigorous probabilistic semantics involves some theoretical legwork, the resulting distance function is actually quite simple - box intersection and volume operations are simply coordinatewise max/min operations, and the calculations for GumbelBox essentially amount to replacing these max/min operations with $\operatorname{logsumexp}$. In fact, the distance function used in our work is arguably simpler in terms of implementation than the one proposed in BoxE, which dynamically changes the penalty depending on box size and whether the point is inside or outside a box. The distance function in our work also has the benefit of being smooth.
> > >
> > > 3.  **Imposing constraints**: You mentioned that subclass constraints such as “C is a subclass of D” can be imposed in the BoxE model by enforcing box containment and that whenever a C-box is contained in a D-box any prediction for class C will be consistent, with many possible configurations possible which adhere to this requirement. These exact same statements apply equally well to our model (see Proposition 3) and in fact have the pleasing probabilistic formulation $P_\text{Box}(\text{Box}(D)\mid\text{Box}(C))=1$. Thus, as stated in our paper, such constraints can be imposed using the same exact distance function (see equation (5)). Furthermore, the probabilistic formulation allows for easy expression and injection of more involved relationships, such as “all items labeled C but not D are labeled E” which is represented as $P_\text{Box}(\text{Box}(E) | \text{Box}(C), \text{Box}(D)^c)=1$, as well as soft relationships such as “50% of items labeled C are labeled D” which is expressed as $P_\text{Box}(\text{Box}(D) | \text{Box}(C))=0.5$, and all these regularization losses would be in the same space as the primary classification loss. In our experiments, we observed that it is extremely important to consider the way in which the constraint loss interacts with the classification loss, as the input $x$ is encoded with a neural network rather than an embedding with free parameters as in KB completion, and thus the fact that the probabilistic formulation allows for constraints which are well-aligned with the primary loss objective is very beneficial.

---

> > > > ### Author Response · Authors · 2021-11-13
> > > > **Few additional points on probabilistic semantics and closing remarks on BoxE as a baseline**
> > > >
> > > > As stated in the previous response titled "Probabilistic semantics and the limited value of adding BoxE as a baseline", while the score from BoxE does not seem to provide any particular benefits, there *are* additional benefits to using a model with valid probabilistic semantics beyond those presented above:
> > > >
> > > > 1. **Strong training loss**: Since $P_\text{MBM}$ is a faithful probability model, one can use negative log-likelihood to train, which tends to converge more quickly and results in a model with less “slack” than a margin-based loss with negative sampling.
> > > >
> > > > 2. **Additional structure**: The proofs of the various propositions in our work are simple precisely because we can leverage the fact that $P_\text{MBM}$ is a faithful probability model.
> > > >
> > > > 3. **Generalizability to non-hierarchical MLC and beyond**: Due to the probabilistic semantics, the proposed method generalizes to any structured prediction task that can be represented as a vector of binary random variables.
> > > >
> > > >  4. **Ability to impose more complex constraints**: As stated above in the “imposing constraints” section, the probabilistic framework allows to impose constraints that involve *multiple* negations/complements in one constraint.
> > > >
> > > >
> > > >
> > > > We agree that a rigorous evaluation of the fundamentals--when to use vector-to-box distance vs. box-to-box, differences in training dynamics for the various score functions, biases and capabilities of each approach--would be an important contribution to this area of research, and is a general question which is relevant for and should be evaluated on all tasks which use box embeddings--not just multi-label classification, but also graph embeddings, knowledge base completion, entailment tasks, etc. While such a comparison would be interesting, it is not in the scope of this paper. For this work, we see very limited value in adding an incomplete comparison for such a general question, i.e., by just using BoxE distance as the representative for all other distance functions that can be used.

---

> > > > > ### Author Response · Authors · 2021-11-23
> > > > > **Adding BoxE as a baseline**
> > > > >
> > > > > Based on your suggestion, we have implemented a non-probabilistic box embedding baseline using the scoring function presented in [Abboud et al. (2020)]([https://proceedings.neurips.cc/paper/2020/hash/6dbbe6abe5f14af882ff977fc3f35501-Abstract.html](https://proceedings.neurips.cc/paper/2020/hash/6dbbe6abe5f14af882ff977fc3f35501-Abstract.html)). The implementation details for this baseline can be found in Section 5 and Appendix A.1 and the performance metrics in Table 1 of the latest revision of the paper.
> > > > >
> > > > > As seen from the results, while BoxE is a strong baseline (better than MHM and C-HMCNN on FUNCAT datasets), MBM performs significantly better than BoxE. Moreover, as seen from Figure 3, while BoxE does capture the label taxonomy to some extent, it does not do it as well as the MBM. As noted in our previous response, we believe that both these are the results of employing a weaker margin-based loss function and heuristically designed box-point distance.
> > > > >
> > > > > We thank you for suggesting an interesting baseline (BoxE), the inclusion of which aids in showing the importance of the probabilistic semantics used in formulating the MBM model.

---

> > > > > > ### Comment · Reviewer_Eo7g · 2021-11-29
> > > > > > **Thanks for the update**
> > > > > >
> > > > > > I didn't quite understand why you initially preferred to dismiss this obvious baseline, but I am glad you agreed in adding a non-probabilistic model as a baseline. Since pretty much any existing box embedding model can be used (i.e., the scoring function of Query2Box could also be used), this is a very important ablation study (originally missed) that could reveal whether the specific choices you set actually make an important difference.
> > > > > >
> > > > > > Unfortunately, the new empirical results are far from clear and appear very misleading: (1) I could not find the margin parameter used for BoxE (which needs to be tuned carefully in a negative sampling loss), and you are using a completely different loss function than the original paper and misrepresenting the work. The loss that you describe in Appendix 1 is simply a different loss and sets margin to 1 by default which would explain the deteriorating performance on some datasets (especially with more classes): Observe that BoxE reports high margin values. It is also not clear whether the original configuration, e.g., tanh mapping, was used.( 2) I do not understand why you did not include BoxE-T, analogously to other models, where you explicitly inject the hierarchy. (3) You argue for the strengths of your approach, e.g., you mention you can encode more complex constraints, but you completely overlook the fact that you can even encode relational constraints using BoxE on top of class constraints (i.e., R(X,Y) and S(Y,X) -> T(X,Y)) which is beyond the approach proposed here.  This paper is not ready for publication.

---

> > > > > > > ### Author Response · Authors · 2021-11-30
> > > > > > > **Regarding the implementation details of BoxE**
> > > > > > >
> > > > > > > Please find our answers to your questions below:
> > > > > > >
> > > > > > > > I didn't quite understand why you initially preferred to dismiss this obvious baseline.
> > > > > > >
> > > > > > > Adapting models which have not previously been applied to MLC takes some care, as you point out with the various choices and tuning required for BoxE. For example, we noticed that Abboud et al. 2020 does not include **any** box embedding baselines - not for probabilistic box embeddings (cited as Vilnis et al. 2018, 2 years prior) nor for Query2Box, which is mentioned explicitly in the related work and which would serve as a trivial adaptation of the score function for their model. The work required to adapt the score function from probabilistic box embeddings on the multi-relational framework proposed in BoxE is similar to the level of work required to adapt BoxE to the task of MLC. We absolutely agree a complete ablation study of all such box embedding models would be of interest, however such comparisons should be made on a wide range of tasks, not simply MLC. In this work, our sole focus is the use of box embeddings for MLC, which itself has not been done before using any existing box embedding method. In our previous response, we have a list of reasons why we believe a probabilistic model is better than a heuristically designed distance function with margin-based loss. We request you point out which points you disagree with so that we can elaborate on them further.
> > > > > > >
> > > > > > >
> > > > > > > > I could not find the margin parameter used for BoxE (which needs to be tuned carefully in a negative sampling loss)
> > > > > > >
> > > > > > > Yes, we did not tune the margin parameter extensively but fixed it to 1. The search ranges provided in BoxE paper have 18 values for the margin, and 3 values for the number of negative samples. This is 18*3*12(datasets)=648 settings. Moreover, the values for the number of negatives has to be changed based on the nature of the MLC dataset. It is non-trivial to tune these parameters. This is another reason why one would prefer a probabilistic model--because it has strong training loss that does not require tuning these parameters. To avoid the cost of tuning of these two hyper-parameters we use a different, but fairly reasonable, margin-based loss compared to the one used in BoxE (please see the next response titled "Final comments on comparison with non-probabilistic box models" for closing remarks on these design choices).
> > > > > > >
> > > > > > > > It is also not clear whether the original configuration, e.g., tanh mapping, was used.
> > > > > > >
> > > > > > > We tried using the tanh mapping. However, it did not work well with our choice of the margin-based loss.
> > > > > > >
> > > > > > > > I do not understand why you did not include BoxE-T, analogously to other models, where you explicitly inject the hierarchy.
> > > > > > >
> > > > > > > Again, when there are 12 datasets involved, it is not trivial to add baselines. Based on the results of BoxE without explicit hierarchy constraints, we believed that running BoxE-T does not justify its training cost.
> > > > > > >
> > > > > > > > You argue for the strengths of your approach, e.g., you mention you can encode more complex constraints, but you completely overlook the fact that you can even encode relational constraints using BoxE on top of class constraints (i.e., R(X,Y) and S(Y,X) -> T(X,Y)) which is beyond the approach proposed here.
> > > > > > >
> > > > > > > Please note that this paper is on multi-label classification where one has only unary relations like $A(x), B(x), \dots$, where $A, B, \dots$ are the labels (as mentioned in your second response). **We are not interested in constraints on non-unary relations $R(X,Y)$ and $S(Y,X) -> T(X,Y)$ as there are no non-unary relations!**. Here we are interested in constraints of the form $A(x) \bigwedge \neg B(x) \bigwedge C(x) \bigwedge \neg D(x) \implies E(x)$. Please see table 1 of the BoxE paper, none of the constraints provided there match this form. But due to the fact that our model is a probabilistic one the constraints involving multiple negations can be incorporated by using $1 - P_{\text{MBM}}(B|x)$.

---

> > > > > > > > ### Author Response · Authors · 2021-11-30
> > > > > > > > **Final comments on comparison with non-probabilistic box models and renaming BoxE baseline.**
> > > > > > > >
> > > > > > > > It has to be noted that box embeddings have not been used for the task of multi-label classification before. There are no "obvious" published baselines that use box embeddings for this task. So the implementation of any box baseline would require careful design decisions and would in itself be a novel contribution. In this paper, we provide theoretical and empirical evidence that the probabilistic box model performs well at the task of hierarchical MLC. It is clear from the paper's presentation and our previous response, which lists several non-probabilistic distance functions, that we do not claim that the probabilistic box model is the only box model that can work well. This is not the intention of the paper. **The paper compares the proposed box model with several strong non-box baselines as well as the state-of-the-art model (C-HMCNN) for the task, showing that the proposed model provides benefits in performance and as well as coherence. It also presents analysis of several interesting geometric aspects of the learned label embeddings. This is a significant and novel contribution that should not be dismissed because it does not compare with a baseline that is not even published for this task!**
> > > > > > > >
> > > > > > > > We believe that checking whether some other non-probabilistic box model works better than the proposed model is not a "baseline comparison" task but should be performed as a follow-up work when one knows, through this work, that some form of box embedding model does work well for MLC. We have provided justification regarding this in our previous responses. **That being said, based on your feedback, we made a good faith effort to provide a non-probabilistic box baseline. Our intention was to just use BoxE's score function with a reasonable margin-based loss as a representative baseline for non-probabilistic box models. However, if you feel that we are misrepresenting the paper by Abboud et. al. by calling this model BoxE, we will change the name of the baseline to "non-probabilistic box model (NPMBM)".**

---

### Decision · Program_Chairs · 2022-01-20

**Decision:**

Accept (Poster)

**Comment:**

The paper studies multi label classification problem. Particularly, they introduce multi-label box model, which uses probabilistic semantics of box embeddings, representing labels as boxes instead of vectors. Their model is evaluated extensively on 12 datasets, and reviewers agreed the paper was well written and well motivated. While it is pretty straightforward application of box embeddings to multi label problem, it is well motivated and the paper adds to the existing literature on box embeddings.

Reviewer Eo7g had a concern with experimental setting, including missing a baseline Abboud et al. (2020). Even after the baseline was added, the reviewer was not convinced about the model’s performance, as the baseline was not extensively tuned. The authors responded with the heavy computational costs of associated with tuning the margin. Reviewer X5cP also pointed the problem with HMC dataset, for which the experimental results should be updated. Given the issues, the paper could benefit from another round of revisions.